# Recovery of sparse linear classifiers from mixture of responses

**Venkata Gandikota**
University of Massachusetts Amherst, MA
Amherst, MA 01003
gandikota.venkata@gmail.com

**Arya Mazumdar**
University of Massachusetts Amherst, MA
Amherst, MA 01003
arya@cs.umass.edu

**Soumyabrata Pal**
University of Massachusetts Amherst, MA
Amherst, MA 01003
soumyabratap@umass.edu

## Abstract

In the problem of learning a *mixture of linear classifiers*, the aim is to learn a collection of hyperplanes from a sequence of binary responses. Each response is a result of querying with a vector and indicates the side of a randomly chosen hyperplane from the collection the query vector belong to. This model provides a rich representation of heterogeneous data with categorical labels and has only been studied in some special settings. We look at a hitherto unstudied problem of query complexity upper bound of recovering all the hyperplanes, especially for the case when the hyperplanes are sparse. This setting is a natural generalization of the extreme quantization problem known as 1-bit compressed sensing. Suppose we have a set of $\ell$ unknown $k$-sparse vectors. We can query the set with another vector $\boldsymbol{a}$, to obtain the sign of the inner product of $\boldsymbol{a}$ and a randomly chosen vector from the $\ell$-set. How many queries are sufficient to identify all the $\ell$ unknown vectors? This question is significantly more challenging than both the basic 1-bit compressed sensing problem (i.e., $\ell = 1$ case) and the analogous regression problem (where the value instead of the sign is provided). We provide rigorous query complexity results (with efficient algorithms) for this problem.

## 1 Introduction

One of the first and most basic tasks of machine learning is to train a binary linear classifier. Given a set of explanatory variables (features) and the binary responses (labels), the objective of this task is to find the hyperplane in the space of features that best separates the variables according to their responses. In this paper, we consider a natural generalization of this problem and model a classification task as a mixture of $\ell$ components. In this generalization, each response is stochastically generated by picking a hyperplane uniformly from the set of $\ell$ unknown hyperplanes, and then returning the side of that hyperplane the feature vector lies. The goal is to learn all of these $\ell$ hyperplanes as accurately as possible, using the least number of responses.

This can be termed as a mixture of binary linear classifiers [33]. Similar mixture of simple machine learning models have been around for at least the last thirty years [9] with mixture of linear regression models being the most studied ones [8, 19, 23, 30, 32, 34, 35, 37]. Models of this type are pretty good function approximators [4, 21] and have numerous applications in modeling heterogeneous settings such as machine translation [25], behavioral health [11], medicine [5], object recognition [28] etc. While algorithms for learning the parameters of mixture of linear regressions are solidly grounded (such as tensor decomposition based learning algorithms of [7]), in many of

the above applications the labels are discrete categorical data, and therefore a mixture of classifiers is a better model than mixture of regressions. To the best of our knowledge, [33] first rigorously studied a mixture of linear classifiers and provided polynomial time algorithm to approximate the subspace spanned by the component classifiers (hyperplane-normals) as well as a prediction algorithm that given a feature and label, correctly predicts the component used. In this paper we study a related but different problem: the sample complexity of learning *all* the component hyperplanes. Our model also differs from [33] where the component responses are 'smoothened out'. Here the term *sample complexity* is used with a slightly generalized meaning than traditional learning theory - as we explain next, and then switch to the term *query complexity* instead.

Recent works on mixture of sparse linear regressions concentrate on an active query based setting [24, 26, 36], where one is allowed to design a sample point and query an oracle with that point. The oracle then randomly chooses one of the component models and returns the answer according to that model. In this paper we adapt exactly this setting for binary classifiers. We assume while queried with a point (vector), an oracle randomly chooses one of the $\ell$ binary classifiers, and then returns an answer according to what was chosen. For the most of this paper we concentrate on recovering 'sparse' linear classifiers, which implies that each of the classifiers uses only few of the explanatory variables. This setting is in spirit of the well-studied *1-bit compressed sensing* (1bCS) problem.

**1-bit compressed sensing.**  In 1-bit compressed sensing, linear measurements of a sparse vector are quantized to only 1 bit, e.g. indicating whether the measurement outcome is positive or not, and the task is to recover the vector up to a prescribed Euclidean error with a minimum number of measurements. An overwhelming majority of the literature focuses on the nonadaptive setting for the problem [1, 2, 14, 17, 20, 27]. Also, a large portion of the literature concentrates on learning only the support of the sparse vector from the 1-bit measurements [1, 17].

It was shown in [20] that $O(\frac{k}{\epsilon} \log(\frac{n}{\epsilon}))$ Gaussian queries[1] suffice to approximately (to the Euclidean precision $\epsilon$) recover an unknown $k$-sparse vector $\boldsymbol{\beta}$ using 1-bit measurements. Given the labels of the query vectors, one recovers $\boldsymbol{\beta}$ by finding a $k$-sparse vector that is consistent with all the labels. If we consider enough queries, then the obtained solution is guaranteed to be close to the actual underlying vector. [1] studied a two-step recovery process, where in the first step, they use queries corresponding to the rows of a special matrix, known as *Robust Union Free Family (RUFF)*, to recover the support of the unknown vector $\boldsymbol{\beta}$ and then use this support information to approximately recover $\boldsymbol{\beta}$ using an additional $\tilde{O}(\frac{k}{\epsilon})$ Gaussian queries. Although the recovery algorithm works in two steps, the queries are nonadaptive.

**Mixture of sparse linear classifiers.**  The main technical difficulty that arises in recovering multiple sparse hyperplanes using 1-bit measurements (labels) is to *align* the responses of different queries concerning a fixed unknown hyperplane. To understand this better, let us consider the case when $\ell = 2$ (see Figure 1). Let $\boldsymbol{\beta^1}, \boldsymbol{\beta^2}$ be two unknown $k$-sparse vectors corresponding to two sparse linear classifiers. On each query, the oracle samples a $\boldsymbol{\beta^i}$, for $i \in \{1, 2\}$, uniformly at random and returns the binary label corresponding to it ($+$ or $-$). One can query the oracle repeatedly with the same query vector to ensure a response from both the classifiers with overwhelmingly high probability.

For any query vector if the responses corresponding to the two classifiers are the same (*i.e.*, $(+, +)$ or $(-, -)$), then we do not gain any information separating the two classifiers. We might still be able to reconstruct some sparse hyperplanes, but the recovery guarantees of such an algorithm will be poor. On the other hand, if both the responses are different (*i.e.*, $(+, -)$), then we do not know which labels correspond to a particular classifier. For example, if the responses are $(+, -)$ and $(+, -)$ for two distinct query vectors, then we do not know if the 'plusses' correspond to the same classifier. This issue of alignment makes the problem challenging. Such alignment issues are less damning in the case of mixture of linear regressions even in the presence of noise [24, 26, 36] since we can utilize the magnitude information of the inner products (labels) to our advantage.

One of the challenges in our study is to recover the supports of the two unknown vectors. Consider the case when $\text{supp}(\boldsymbol{\beta^1}) \neq \text{supp}(\boldsymbol{\beta^2})$, where $\text{supp}(\boldsymbol{v})$ denotes the support of the vector $\boldsymbol{v} \in \mathbb{R}^n$. In this case, we show that using an RUFF in combination with another similar class of union-free family (UFF), we can deduce the supports of both $\boldsymbol{\beta^1}$ and $\boldsymbol{\beta^2}$. Wielding known constructions of

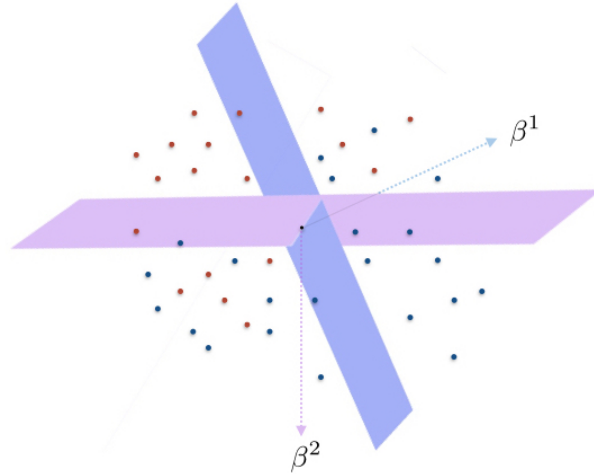

Figure 1: Recover the two lines given red triangles and blue dots. How many such points do we require in order to recover the two lines?

such UFFs from literature, we can recover the supports of both $k$-sparse vectors using $O(k^3 \log^2 n)$ queries. Once we obtain the supports, we use an additional $O(\frac{k}{\epsilon} \log nk)$ Gaussian queries (with a slight modification) to approximately recover the individual vectors.

We then extend this two-step process (using more general classes of UFFs) to recover a mixture of $\ell$ different sparse vectors under the assumption that the support of no vector is contained in the union of supports of the remaining ones (Assumption 1). The assumption implies that if the sparse vectors are arranged as columns of a matrix, then the matrix contains the identity matrix as a permutation of the rows. This separability condition appears before in [3, 12, 31] in the context of nonnegative integer matrix factorization, which is a key tool that we will subsequently use to prove our results. To quote [3] in the context of matrix factorization, "an approximate separability condition is regarded as a fairly benign assumption and is believed to hold in many practical contexts in machine learning." We believe this observation holds for our context as well (each classifier uses some unique feature).

We show that with this support separability condition, $\tilde{O}(\ell^6 k^3)$ queries suffice for support recovery of $\ell$ different $k$-sparse vectors. Further, using $\tilde{O}((\ell^3 k/\epsilon))$ queries, we can recover each of the $\boldsymbol{\beta^i}$'s, for $i \in \{1, \ldots, \ell\}$ up to $\epsilon$ precision (see Theorem 1 and Theorem 2).

The two-stage procedure described above, can be made completely non-adaptive using queries from union free families (see Theorem 3).

Furthermore, for $\ell = 2$, we see that the support condition (Assumption 1) is not necessary. We can approximately recover the two unknown vectors provided a) they are not extremely sparse and, b) each $\boldsymbol{\beta^i} \in \delta \mathbb{Z}^n$ for some $\delta > 0$. To prove this, we borrow the tools from [2] who give guarantees for 1-bit compressed sensing using sub-Gaussian vectors. In particular, we use queries with independent Bernoulli coordinates which are sub-Gaussian. These discrete random queries (as opposed to continuous Gaussians) along with condition (b), enables us to align the labels corresponding to the two unknown vectors. (see Theorem 4 for more details). Note that condition (a) is due to the result by [2] and is necessary for recovery using sub-Gaussian queries and (b) is a mild assumption on the precision of the unknown vectors, which was also necessary [24, 36] for learning the mixture of sparse linear regressions.

**Technical take-away.** As stated above, the main technical hurdle in the paper lies in the support recovery problem for the sparse vectors. We do it in a few (non-adaptive) steps. First, we try to design queries that will lead us to estimate the size of the intersections of supports for every pair of sparse vectors. If we can design such a set of queries, then using a nonnegative integer matrix factorization techniques we can estimate all the supports. Indeed, this is where the separability assumption comes in handy. Further, because of the separability assumption, it is also possible to design queries from which we can simulate the response of every unknown vector with a random Gaussian query

and tag it. This allows us to recover and *cluster* the responses of every unknown vector to a set of random Gaussian queries from which we can approximately recover all the unknown vectors.

To estimate the size of the intersections of supports for the vector-pairs, we rely on combinatorial designs (and related set-systems), such as pairwise independent cover free families [15]. While some such union-free families have been used to estimate the support in 1-bit compressed sensing before [1], the use of pairwise independent sets to *untangle multiple sparse vectors* is new and has almost nothing to do with the recovery of sparse vectors itself.

We leave the problem of designing a query scheme that works for any general $\ell$ without any assumptions as an open problem. Lack of Assumption 1 seems to be a fundamental barrier to support recovery as it ensures that a sparse vector will never be in the span of the others. However, a formal statement of this effect still eludes us. For large $\ell$, finding out the dependence of query complexity on $\ell$ is also a natural question. Overall, this study leads to an interesting set of questions that are technically demanding as well as quite relevant to practical modeling of heterogeneous data that are ubiquitous in applications. For instance, in recommendation systems, where the goal is to identify the factors governing the preferences of individual members of a group via crowdsourcing while preserving the anonymity of their responses.

**Organization.** The rest of this paper is organized as follows. In the next section, we formally define the problem statement followed by a list of our contributions in Section 3. The notations and the necessary background on various families of sets are presented in Section 4. We prove Theorem 1 in Section 5. The proofs of all the other theorems and the helper lemmas required for the proof of Theorem 1 are deferred to the supplementary material. Moreover, we demonstrate the capability of our algorithms to learn movie genre preferences of two unknown users using the MovieLens [18] dataset. The experimental details are included in Section E of the supplementary material.

## 2 Problem Statement

Let $\boldsymbol{\beta}^1, \boldsymbol{\beta}^2, \ldots, \boldsymbol{\beta}^\ell \in \mathbb{R}^n$ be a set of $\ell$ unknown $k$-sparse vectors. Each $\boldsymbol{\beta}^i$ defines a linear classifier that assigns a label from $\{-1, 0, 1\}$ to every vector in $\boldsymbol{v} \in \mathbb{R}^n$ according to $\mathsf{sign}(\langle \boldsymbol{v}, \boldsymbol{\beta} \rangle)$, where the sign function $\mathsf{sign} : \mathbb{R} \to \{-1, 0, 1\}$, is defined as,

$$\mathsf{sign}(x) = \begin{cases} +1 & \text{if } x > 0 \\ -1 & \text{if } x < 0 \\ 0 & \text{if } x = 0 \end{cases}.$$

**Remark 1.** *It is also possible to handle binary output from the classifier instead of ternary as has been studied in this paper. See the full version [16] for more details.*

As described above, our work focuses on recovering the unknown classifiers in the *query model* that was used in [36, 24] to study the mixtures of sparse linear regressions. In the query model, we assume the existence of an oracle $\mathcal{O}$ which when queried with a vector $\boldsymbol{v} \in \mathbb{R}^n$, samples one of the classifiers $\boldsymbol{\beta} \in \{\boldsymbol{\beta}^1, \boldsymbol{\beta}^2, \ldots, \boldsymbol{\beta}^\ell\}$ uniformly at random and returns the label of $\boldsymbol{v}$ assigned by the sampled classifier $\boldsymbol{\beta}$. The goal of approximate recovery is to reconstruct each of the unknown classifiers using small number of oracle queries. The problem can be formalized as follows:

**Problem 1** ($\epsilon$-recovery). *Given $\epsilon > 0$, and query access to oracle $\mathcal{O}$, find $k$-sparse vectors $\{\hat{\boldsymbol{\beta}}^1, \hat{\boldsymbol{\beta}}^2, \ldots, \hat{\boldsymbol{\beta}}^\ell\}$ such that for some permutation $\sigma : [\ell] \to [\ell]$*

$$\left\| \frac{\boldsymbol{\beta}^i}{\|\boldsymbol{\beta}^i\|_2} - \frac{\hat{\boldsymbol{\beta}}^{\sigma(i)}}{\|\hat{\boldsymbol{\beta}}^{\sigma(i)}\|_2} \right\|_2 \leq \epsilon \quad \forall i \in [\ell].$$

Since from the classification labels, we lose the magnitude information of the unknown vectors, we assume each $\boldsymbol{\beta}^i$ and the estimates $\hat{\boldsymbol{\beta}}^i$ to have a unit norm.

Similar to the literature on one-bit compressed sensing, one of our proposed solutions employs a two-stage algorithm to recover the unknown vectors. In the first stage the algorithm recovers the support of every vector, and then in the second stage, approximately recovers the vectors using the support information.

For any vector $\boldsymbol{v} \in \mathbb{R}^n$, let $\mathsf{supp}(\boldsymbol{v}) := \{i \in [n] \mid \boldsymbol{v}_i \neq 0\}$ denote the support of $\boldsymbol{v}$. The problem of support recovery is then defined as follows:

**Problem 2** (Support Recovery). *Given query access to oracle $\mathcal{O}$, construct $\{\hat{\boldsymbol{\beta}}^1, \hat{\boldsymbol{\beta}}^2, \ldots, \hat{\boldsymbol{\beta}}^\ell\}$ such that for some permutation $\sigma : [\ell] \rightarrow [\ell]$*

$$\mathsf{supp}(\hat{\boldsymbol{\beta}}^{\boldsymbol{\sigma(i)}}) = \mathsf{supp}(\boldsymbol{\beta}^{\boldsymbol{i}}) \quad \forall \, i \in [\ell]$$

For both these problems, we primarily focus on minimizing the query complexity of the problem, *i.e.*, minimizing the number of queries that suffice to approximately recover all the sparse unknown vectors or their supports. However, all the algorithms proposed in this work also run in time $O(\mathsf{poly}(q))$, where $q$ is the query complexity of the algorithm.

# 3 Our contributions

In order to present our first set of results, we need certain assumption regarding the separability of supports of the unknown vectors. In particular, we want each component of the mixture to have a unique identifying coordinate. More formally, it can be stated as follows:

**Assumption 1.** *For every $i \in [\ell]$, $\mathsf{supp}(\boldsymbol{\beta}^{\boldsymbol{i}}) \not\subseteq \bigcup_{j:j\neq i} \mathsf{supp}(\boldsymbol{\beta}^{\boldsymbol{j}})$, i.e. the support of any unknown vector is not contained in the union of the support of the other unknown vectors.*

**Two-stage algorithm:** First, we propose a two-stage algorithm for $\epsilon$-recovery of the unknown vectors. In the first stage of the algorithm, we recover the support of the unknown vectors (Theorem 1), followed by $\epsilon$-recovery using the deduced supports (Theorem 2) in the second stage. Each stage in itself is non-adaptive, *i.e.*, the queries do not depend on the responses of previously made queries.

**Theorem 1.** *Let $\{\boldsymbol{\beta}^1, \ldots, \boldsymbol{\beta}^\ell\}$ be a set of $\ell$ unknown $k$-sparse vectors in $\mathbb{R}^n$ that satisfy Assumption 1. There exists an algorithm to recover the support of every unknown vector $\{\boldsymbol{\beta}^{\boldsymbol{i}}\}_{i\in[\ell]}$ with probability at least $1 - O(1/n^2)$, using $O(\ell^6 k^3 \log^2 n)$ non-adaptive queries to oracle $\mathcal{O}$.*

Now using this support information, we can approximately recover the unknown vectors using an additional $\tilde{O}(\ell^3 k)$ non-adaptive queries.

**Theorem 2.** *Let $\{\boldsymbol{\beta}^1, \ldots, \boldsymbol{\beta}^\ell\}$ be a set of $\ell$ unknown $k$-sparse vectors in $\mathbb{R}^n$ that satisfy Assumption 1. There exists a two-stage algorithm that uses $O\left(\ell^6 k^3 \log^2 n + (\ell^3 k/\epsilon) \log(nk/\epsilon) \log(k/\epsilon)\right)$ oracle queries for the $\epsilon$-recovery of all the unknown vectors with probability at least $1 - O(1/n)$.*

**Remark 2.** *We note that for the two-stage recovery algorithm to be efficient, we require the magnitude of non-zero entries of the unknown vectors to be non-negligible (at least $1/exp(n)$). This assumption however is not required to bound the query complexity of the algorithm which is the main focus of this work.*

**Completely non-adaptive algorithm:** Next, we show that the entire $\epsilon$-recovery algorithm can be made non-adaptive (single-stage) at the cost of increased query complexity.

**Theorem 3.** *Let $\{\boldsymbol{\beta}^1, \ldots, \boldsymbol{\beta}^\ell\}$ be a set of $\ell$ unknown $k$-sparse vectors in $\mathbb{R}^n$ that satisfy Assumption 1. There exists an algorithm that uses $O\left((\ell^{\ell+3} k^{\ell+2}/\epsilon) \log n \log(n/\epsilon) \log(k/\epsilon)\right)$ non-adaptive oracle queries for the $\epsilon$-recovery of all the unknown vectors with probability at least $1 - O(1/n)$.*

Note that even though the one-stage algorithm uses many more queries than the two-stage algorithm, a completely non-adaptive is highly parallelizable as one can choose all the query vectors in advance. Also, in the $\ell = O(1)$ regime, the query complexity is comparable to its two-stage analogue.

While we mainly focus on minimizing the query complexity, all the algorithms proposed in this work run in $\mathsf{poly}(n)$ time assuming every oracle query takes $\mathsf{poly}(n)$ time and $\ell = o(\log n)$.

**Non-adaptive algorithm for $\ell = 2$ without Assumption 1:** For $\ell = 2$, we do not need the separability condition (Assumption 1) required earlier for support recovery. Even for $\epsilon$-recovery, instead of Assumption 1, we just need a mild assumption on the precision $\delta$, and the sparsity of the unknown vectors. In particular, we propose an algorithm for the $\epsilon$-recovery of the two unknown vectors using $\tilde{O}(k^3 + k/\epsilon)$ queries provided the unknown vectors have some finite precision and are not extremely sparse.

**Assumption 2.** *For $\boldsymbol{\beta} \in \{\boldsymbol{\beta}^1, \boldsymbol{\beta}^2\}$, $\|\boldsymbol{\beta}\|_\infty = o(1)$.*

Assumption 2 ensures that we can safely invoke the result of [2] who use the exact same assumption in the context of 1-bit compressed sensing using sub-Gaussian queries.

**Theorem 4.** *Let $\beta^1, \beta^2$ be two $k$-sparse vectors in $\mathbb{R}^n$ that satisfy Assumption 2. Let $\delta > 0$ be the largest real such that $\beta^1, \beta^2 \in \delta\mathbb{Z}^n$. There exists an algorithm that uses $O(k^3 \log^2 n + (k^2/\epsilon^4\delta^2) \log^2(n/k\delta^2))$ (adaptive) oracle queries for the $\epsilon$-recovery of $\beta^1, \beta^2$ with probability at least $1 - O(1/n)$.*

*Moreover, if $\mathsf{supp}(\beta^1) \neq \mathsf{supp}(\beta^2)$, then there exists a two-stage algorithm for the $\epsilon$-recovery of the two vectors using only $O(k^3 \log^2 n + (k/\epsilon) \log(nk/\epsilon) \log(k/\epsilon))$ non-adaptive oracle queries.*

Also, the $\epsilon$-recovery algorithm proposed for Theorem 4 runs in time $\mathsf{poly}(n, 1/\delta)$.

**No sparsity constraint:** We can infact avoid the sparsity constraint altogether for the case of $\ell = 2$. Since in this setting, we consider the support of both unknown vectors to include all coordinates, we do not need a support recovery stage. We then get a single stage and therefore completely non-adaptive algorithm for $\epsilon$-recovery of the two unknown vectors.

**Corollary 1.** *Let $\beta^1, \beta^2$ be two unknown vectors in $\mathbb{R}^n$ that satisfy Assumption 2. Let $\delta > 0$ be the largest real such that $\beta^1, \beta^2 \in \delta\mathbb{Z}^n$. There exists an algorithm that uses $O((n^2/\epsilon^4\delta^2) \log(1/\delta))$ non-adaptive oracle queries for the $\epsilon$-recovery of $\beta^1, \beta^2$ with probability at least $1 - O(1/n)$.*

# 4 Preliminaries

Let $[n]$ to denote the set $\{1, 2, \ldots, n\}$. For any vector $v \in \mathbb{R}^n$, $\mathsf{supp}(v)$ denotes the support and $v_i$ denote the $i^{th}$ entry (coordinate) of the vector $v$. We will use $e_i$ to denote a vector which has 1 only in the $i^{th}$ position and is 0 everywhere else. We will use the notation $\langle a, b \rangle$ to denote the inner product between two vectors $a$ and $b$ of the same dimension. For a matrix $A \in \mathbb{R}^{m \times n}$, let $A_i \in \mathbb{R}^n$ be its $i^{th}$ column and $A[j]$ denote its $j^{th}$ row. and let $A_{i,j}$ be the $(i, j)$-th entry of $A$. We will denote by $\mathsf{Inf}$ a *very large positive number*. Also, let $\mathcal{N}(0, 1)$ denote the standard normal distribution. We will use $\mathcal{P}_n$ to denote a the set of all $n \times n$ permutation matrices, *i.e.*, the set of all $n \times n$ binary matrices that are obtained by permuting the rows of an $n \times n$ identity matrix (denoted by $I_n$). Let $\mathsf{round} : \mathbb{R} \to \mathbb{Z}$ denote a function that returns the closest integer to a given real input.

Let us further introduce a few definitions that will be used throughout the paper.

**Definition 3.** *For a particular entry $i \in [n]$, define $\mathcal{S}(i)$ to be the set of all unknown vectors whose $i^{th}$ entry is non-zero.*

$$\mathcal{S}(i) := \{\beta^j, j \in [\ell] \mid \beta^j_{\ i} \neq 0\}$$

**Definition 4.** *For a particular query vector $v$, define $\mathsf{poscount}(v)$, $\mathsf{negcount}(v)$ and $\mathsf{nzcount}(v)$ to be the number of unknown vectors that assign a positive, negative, and non-zero label to $v$ respectively.*

$$\mathsf{poscount}(v) := |\{\beta^j \mid \mathsf{sign}(\langle v, \beta^j \rangle) = +1, j \in [\ell]\}|$$
$$\mathsf{negcount}(v) := |\{\beta^j \mid \mathsf{sign}(\langle v, \beta^j \rangle) = -1, j \in [\ell]\}|$$
$$\mathsf{nzcount}(v) := \mathsf{poscount}(v) + \mathsf{negcount}(v)$$
$$= |\{\beta^j \mid \mathsf{sign}(\langle v, \beta^j \rangle) \neq 0, j \in [\ell]\}|.$$

**Definition 5** (Gaussian query). *A vector $v \in \mathbb{R}^n$ is called a Gaussian query vector if each entry $v_i$ of $v$ is sampled independently from the standard Normal distribution, $\mathcal{N}(0, 1)$.*

## 4.1 Estimating the counts

In this section we show how to accurately estimate each of the counts *i.e.*, $\mathsf{poscount}(v)$, $\mathsf{negcount}(v)$ and $\mathsf{nzcount}(v)$ with respect to any query vector $v$, with high probability (see Algorithm 1).

The idea is to simply query the oracle with the same query vector repeatedly and estimate the counts empirically using the responses of the oracle. Let $T$ denote the number of times a fixed query vector $v$ is repeatedly queried. We refer to this quantity as the *batchsize*. We now show that the empirical estimates of each of the counts equals the real counts with high probability.

**Lemma 6.** *For any query vector $v$, Algorithm 1 with batchsize $T$ provides the correct estimates of $\mathsf{poscount}(v)$, $\mathsf{negcount}(v)$ and $\mathsf{nzcount}(v)$ with probability at least $1 - 8e^{-T/2\ell^2}$.*

---

**Algorithm 1** QUERY($\boldsymbol{v}, T$)

---

**Require:** Query access to oracle $\mathcal{O}$.
 1: **for** $i = 1, 2, \ldots, T$ **do**
 2:     Query the oracle with vector $\boldsymbol{v}$ and obtain response $y^i \in \{-1, 0, +1\}$.
 3: **end for**
 4: Let $\hat{\text{pos}} := \text{round}\left(\frac{\ell \sum_i \mathbb{1}[y^i = +1]}{T}\right)$
 5: Let $\hat{\text{neg}} := \text{round}\left(\frac{\ell \sum_i \mathbb{1}[y^i = -1]}{T}\right)$
 6: Let $\hat{\text{nz}} := \hat{\text{pos}} + \hat{\text{neg}}$.
 7: Return $\hat{\text{pos}}, \hat{\text{neg}}, \hat{\text{nz}}$.

---

## 4.2 Family of sets

We now review literature on some important families of sets called *union free families* [1] and *cover free families* [22] that found applications in cryptography, group testing and 1-bit compressed sensing. These special families of sets are used crucially in this work to design the query vectors for the support recovery and the $\epsilon$-recovery algorithms.

**Definition 7** (Robust Union Free Family $(d, t, \alpha) - \mathsf{RUFF}$). *Let $d, t$ be integers and $0 \le \alpha \le 1$. A family of sets, $\mathcal{F} = \{\mathcal{H}_1, \mathcal{H}_2, \ldots, \mathcal{H}_n\}$ where each $\mathcal{H}_i \subseteq [m]$ and $|\mathcal{H}| = d$ is a $(d, t, \alpha)$-$\mathsf{RUFF}$ if for any set of $t$ indices $T \subset [n], |T| = t$, and any index $j \notin T$,*

$$\left| \mathcal{H}_j \setminus \left( \bigcup_{i \in T} \mathcal{H}_i \right) \right| > (1 - \alpha)d.$$

We refer to $n$ as the size of the family of sets, and $m$ to be the alphabet over which the sets are defined. RUFFs were studied earlier in the context of support recovery of 1bCS [1], and a simple randomized construction of $(d, t, \alpha)$-RUFF with $m = O(t^2 \log n)$ was proposed by De Wolf [10].

**Lemma 8.** *[1, 10] Given $n, t$ and $\alpha > 0$, there exists an $(d, t, \alpha)$-$\mathsf{RUFF}$, $\mathcal{F}$ with $m = O\big((t^2 \log n)/\alpha^2\big)$ and $d = O((t \log n)/\alpha)$.*

RUFF is a generalization of the family of sets known as the Union Free Familes (UFF) - which are essentially $(d, t, 1)$-RUFF. In this work, we require yet another generalization of UFF known as Cover Free Families (CFF) that are also sometimes referred to as superimposed codes [13].

**Definition 9** (Cover Free Family $(r, t)$-CFF). *A family of sets $\mathcal{F} = \{\mathcal{H}_1, \mathcal{H}_2, \ldots, \mathcal{H}_n\}$ where each $\mathcal{H}_i \subseteq [m]$ is an $(r, t)$-$\mathsf{CFF}$ if for any pair of disjoint sets of indices $T_1, T_2 \subset [n]$ such that $|T_1| = r, |T_2| = t, T_1 \cap T_2 = \emptyset$,*

$$\left| \bigcap_{i \in T_1} \mathcal{H}_i \setminus \bigcup_{i \in T_2} \mathcal{H}_i \right| > 0.$$

Several constructions and bounds on existence of CFFs are known in literature. We state the following lemma regarding the existence of CFF which can be found in [29, 15]. We also include a proof in the supplementary material for the sake of completeness.

**Lemma 10.** *For any given integers $r, t$, there exists an $(r, t)$-$\mathsf{CFF}$, $\mathcal{F}$ of size $n$ with $m = O(t^{r+1} \log n)$.*

Note that $(1, t)$-CFF is exactly a UFF. The $(2, t)$-CFF is of particular interest to us and will henceforth be referred to as the *pairwise union free family* (PUFF). From Lemma 10 we know the existence of PUFF of size $n$ with $m = O(t^3 \log n)$.

**Corollary 2.** *For any given integer $t$, there exists a $(2, t)$-$\mathsf{CFF}$, $\mathcal{F}$ of size $n$ with $m = O(t^3 \log n)$.*

## 5   Support Recovery (Proof of Theorem 1)

In this section, we present an efficient algorithm to recover the support of all the $\ell$ unknown vectors using a small number of oracle queries. The proof of Theorem 1 follows from the guarantees of Algorithm 2. The proofs of the helper lemmas used in this theorem are deferred to Section B in the supplementary material.

Consider the support matrix $\boldsymbol{X} \in \{0,1\}^{n \times \ell}$ where the $i$-th column is the indicator of $\mathsf{supp}(\boldsymbol{\beta^i})$. The goal in Theorem 1 is to recover this unknown matrix $\boldsymbol{X}$ (up to permutations of columns) using a small number of oracle queries. In Algorithm 2, we recover $\boldsymbol{X}$ from $\boldsymbol{X}\boldsymbol{X^T}$, where the latter can be constructed using only the estimates of nzcount for some specially designed queries. The unknown matrix $\boldsymbol{X}$ is recovered from the constructed $\boldsymbol{X}\boldsymbol{X^T}$ by rank factorization with binary constraints. The factorization is efficient and also turns out to be unique (up to permutations of columns) because of the separability assumption (Assumption 1) on the supports of the unknown vectors.

The main challenge lies in constructing the matrix $\boldsymbol{X}\boldsymbol{X^T}$ using just the oracle queries. Recall that for any $i \in [n]$, $\mathcal{S}(i)$ denotes the set of unknown vectors that have a non-zero entry in the $i$-th coordinate. Note that the $i$-th row of $\boldsymbol{X}$, for any $i \in [n]$, is essentially the indicator of $\mathcal{S}(i)$. From this observation, it follows that the $(i,j)$-th entry of $\boldsymbol{X}\boldsymbol{X^T}$ is captured by the term $|\mathcal{S}(i) \cap \mathcal{S}(j)|$.

We observe that the quantity $|\mathcal{S}(i) \cap \mathcal{S}(j)|$ can be computed from oracle queries in two steps. First, we use query vectors from an RUFF with appropriate parameters to compute $|\mathcal{S}(i)|$ for every $i \in [n]$ (see Algorithm 3). Then, using queries from a PUFF (Algorithm 4) to obtain $|\mathcal{S}(i) \cup \mathcal{S}(j)|$ for every pair $(i,j)$. To state it formally,

**Lemma 11.** *There exists an algorithm to compute $|\mathcal{S}(i)|$ for each $i \in [n]$ with probability at least $1 - O\left(1/n^2\right)$ using $O(\ell^4 k^2 \log(\ell k n) \log n)$ oracle queries.*

**Lemma 12.** *There exists an algorithm to compute $|\mathcal{S}(i) \cup \mathcal{S}(j)|$ for every pair $(i,j)$ with probability at least $1 - O\left(1/n^2\right)$ using $O(\ell^6 k^3 \log(\ell k n) \log n)$ oracle queries.*

By combining these two steps, we can obtain the $(i,j)$-th entry of $XX^T$ as $|\mathcal{S}(i) \cap \mathcal{S}(j)| = |\mathcal{S}(i)| + |\mathcal{S}(j)| - |\mathcal{S}(i) \cup \mathcal{S}(j)|$. Equipped with these two Lemmas, we now prove the guarantees of Algorithm 2 that completes the proof of Theorem 1.

---

**Algorithm 2** RECOVER–SUPPORT

---

**Require:** Query access to oracle $\mathcal{O}$.
**Require:** Assumption 1 to be true.
 1: Estimate $|S(i)|$ for every $i \in [n]$ using Algorithm 3.
 2: Estimate $|S(i) \cup S(j)|$ for every $i, j \in [n]$ using Algorithm 4.
 3: **for** every pair $(i,j) \in [n] \times [n]$ **do**
 4:     Set $Z_{i,j} = |\mathcal{S}(i)| + |\mathcal{S}(j)| - |\mathcal{S}(i) \cup \mathcal{S}(j)|$
 5: **end for**
 6: Return $\hat{\boldsymbol{X}} \in \{0,1\}^{n \times \ell}$ such that $\hat{\boldsymbol{X}}\hat{\boldsymbol{X}}^T = \boldsymbol{Z}$.

---

*Proof of Theorem 1.* Using Algorithm 3 and Algorithm 4, we compute $|\mathcal{S}(i) \cap \mathcal{S}(j)| = |\mathcal{S}(i)| + |\mathcal{S}(i)| - |\mathcal{S}(i) \cup \mathcal{S}(j)|$ for every pair $(i,j) \in [n] \times [n]$, and hence populate the entries of $Z = XX^T$. To obtain $X$ from $Z$, we perform a rank factorization of $Z$ with a binary constraint on the factors. We now show that Assumption 1 ensures that this factorization is unique up to permutations.

Suppose $\boldsymbol{Y} \neq \boldsymbol{X}$ is a binary matrix such that $\boldsymbol{Y}\boldsymbol{Y}^T = \boldsymbol{X}\boldsymbol{X}^T$. Therefore, there exists a rotation matrix $\boldsymbol{R} \in \mathbb{R}^{\ell \times \ell}$ such that $\boldsymbol{Y} = \boldsymbol{X}\boldsymbol{R}$. From Assumption 1 we know that there exists an $\ell \times \ell$ submatrix $\tilde{\boldsymbol{X}}$ of $\boldsymbol{X}$ that is a permutation matrix. For the corresponding submatrix $\tilde{\boldsymbol{Y}}$ of $\boldsymbol{Y}$ (obtained by choosing the same subset of rows), it must hold that

$$\tilde{\boldsymbol{Y}}\tilde{\boldsymbol{Y}}^T = \tilde{\boldsymbol{X}}\tilde{\boldsymbol{X}}^T = \boldsymbol{I}$$

where $\boldsymbol{I}$ is the $\ell \times \ell$ identity matrix. Since $\boldsymbol{Y}$ has binary entries, $\tilde{\boldsymbol{Y}}$ must be a permutation matrix as well. This implies that $\boldsymbol{R}$ is a permutation matrix and a constrained rank factorization can recover $\boldsymbol{X}$ up to a permutation of columns. Therefore, Algorithm 2 successfully recovers the support of all the $\ell$ unknown vectors.

The total number of queries needed by Algorithm 2 is the sum total of the queries needed by Algorithm 3 and Algorithm 4 which is $O(\ell^6 k^3 \log(\ell k n) \log(n))$.

Moreover, since Algorithm 3 and Algorithm 4 each succeed with probability at least $1 - O(1/n^2)$. By a union bound, it follows that Algorithm 2 succeeds with probability at least $1 - O(1/n^2)$.  □

# 6 Conclusion and Open Questions

In this work, we initiated the study of recovering a mixture of $\ell$ different sparse linear classifiers given query access to an oracle. The problem generalizes the well-studied work on 1-bit compressed sensing ($\ell = 1$) and also complements the literature on learning mixtures of sparse linear regression in a similar query model.

Our results for $\ell > 2$, rely on the assumption that the supports of all the unknown vectors are separable. This separability assumption translates to each classifier using a unique feature not being used in others, which happen often in practice. The approximate recovery problem without the separability assumption is non-trivial even for $\ell = 2$ case, for which we provide guarantees with much milder assumptions on the precision of the classifiers. We leave the problem of support recovery and $\epsilon$-recovery without any assumptions as an open problem.

We primarily focus on providing upper bounds on the query complexity of the support recovery and approximate recovery of the unknown vectors. However, proving optimality results for any such recovery is an interesting open direction. It is known that even to recover the support of a single $k$-sparse vector in the 1-bit compressed sensing setting, about $\Omega(k^2 \log n)$ queries are required. This corresponds to $\ell = 1$ case, and the lower bound holds trivially for any general $\ell$ as well. However, a nontrivial lower bound on the query complexity characterizing the asymptotic dependence on $\ell$, the number of components, will be of interest.

## Broader Impact

This paper is a theoretical study that brings together two seemingly disjoint but equally impactful fields of sparse recovery and mixture models: the first having numerous applications in signal processing while the second being the main statistical model for clustering. Given that, this work belongs to the foundational area of data science and enhances our understanding of some basic theoretical questions. We feel the methodology developed in this paper is instructive, and exemplifies the use of several combinatorial objects and techniques in signal recovery and classification, that are hitherto underused. Therefore we foresee the technical content of this paper to form good teaching material in foundational data science and signal processing courses. The content of this paper can raise interest of students or young researchers in discrete mathematics to applications areas and problems of signal processing and machine learning.

While primarily of theoretical interest, the results of the paper can be immediately applicable to some real-life scenarios and be useful in recommendation systems, one of the major drivers of data science research. In particular, if in any case of feedback/rating from users of a service there is ambiguity about the source of the feedback, our framework can be used. This is also applicable to crowdsourcing applications.

*Acknowledgements:* This research is supported in part by NSF CCF 1909046 and NSF 1934846.

## Footnotes

[1]all coordinates of the query vector are sampled independently from the standard Gaussian distribution

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
