[Supplementary Material]

# Recovery of sparse linear classifiers from mixture of responses
## Supplementary Material

## A   Missing Proofs from Section 4

*Proof of Lemma 5.* The proof of the lemma follows from a simple application of Chernoff bound.

Let $\{y^i\}_{i=1}^T$ be the set of responses obtained by querying the oracle repeatedly $T$ times with vector $\boldsymbol{v}$. Let $Z = \sum_i \mathbb{1}[y^i = +1]$, and therefore $\mathbb{E}Z = \frac{T \times \mathsf{poscount}}{\ell}$.

Note that Algorithm 1 makes a mistake in estimating poscount only if

$$|Z - \frac{T \times \mathsf{poscount}}{\ell}| \geq \frac{T}{2\ell}.$$

Since the responses in each batch are independent, using Chernoff bound [6], we get an upper bound on the probability that Algorithm 1 makes a mistake in estimating poscount as

$$\Pr\left(|Z - \mathbb{E}Z| \geq \frac{T}{2\ell}\right) \leq 2e^{-\frac{T}{2\ell^2}}.$$

The same argument and conclusion holds for observing the negcount of the query vector as well. Also, since $\mathsf{nzcount} = T - (\mathsf{poscount} + \mathsf{negcount})$, using union bound, it follows that $\hat{\mathsf{nz}} \neq \mathsf{nzcount}$ with probability at most $4e^{-\frac{T}{2\ell^2}}$. $\qquad\square$

*Proof of Lemma 9.* We give a non-constructive proof for the existence of $(r,t) - \mathsf{CFF}$ of size $n$ and alphabet $m = O(t^{r+1} \log n)$. Recall that a family of sets $\mathcal{F} = \{\mathcal{H}_1, \mathcal{H}_2, \ldots, \mathcal{H}_n\}$ where each $\mathcal{H}_i \subseteq [m]$ is an $(r,t) - \mathsf{CFF}$ if the following holds: for all distinct $j_0, j_1, \ldots, j_{t+r-1} \in [n]$, it is the case that

$$\bigcap_{p \in \{0,1,\ldots,r-1\}} \mathcal{H}_{j_p} \not\subseteq \bigcup_{q \in \{r, r+1, \ldots, t+r-1\}} \mathcal{H}_{j_q}.$$

Since PUFF is a special case of $(r,t) - \mathsf{CFF}$ for $r = 2$, this result holds for PUFF as well.

Consider a matrix $\boldsymbol{G}$ of size $m \times n$ where each entry is generated independently from a Bernoulli$(p)$ distribution with $p$ as a parameter. Consider a distinct set of $t + r$ indices $j_0, j_1, \ldots, j_{t+1}, \ldots, j_{k+r-1} \in [n]$. For a particular row of the matrix $\boldsymbol{G}$, the event that there exists a 1 in the indices $j_0, j_1, \ldots, j_{r-1}$ and 0 in the indices $j_r, j_{r+1}, \ldots, j_{t+r-1}$ holds with probability $p^r(1-p)^t$. Therefore, for a fixed row, this event does not hold with probability $1 - p^r(1-p)^t$ and the probability that for all the rows the event does not hold is $(1 - p^r(1-p)^t)^m$. Notice that the number of such possible sets of $t + r$ columns is $\binom{n}{t+r}\binom{t+r}{r}$. By taking a union bound, the probability $(P_e)$ that the event does not hold for all the rows for at least one set of $t + r$ indices is

$$P_e \leq \binom{n}{t+r}\binom{t+r}{r}\left(1 - p^r(1-p)^t\right)^m$$

Since we want to minimize the upper bound, we want to maximize $p^r(1-p)^t$. Substituting $p = \frac{1}{t+1}$, we get that

$$p^r(1-p)^t = \left(\frac{t}{t+1}\right)^t \cdot \frac{1}{(t+1)^r} > \frac{1}{e(t+1)^r}.$$

Further, using the fact that $\binom{n}{t} \leq \left(\frac{en}{t}\right)^t$, we obtain

$$P_e \leq \frac{(en)^{t+r}}{(t+r)^t}\left(1 - \frac{1}{e(t+1)^r}\right)^m \leq \frac{(en)^{t+r}}{(t+r)^t}\exp\left(-\frac{m}{e(t+1)^r}\right) < \alpha$$

for some very small number $\eta$. Taking log on both sides and after some rearrangement, we obtain

$$m > e(t+1)^r\left((t+r)\log\frac{en}{t+r} + r\log(t+r) + \log\frac{1}{\eta}\right).$$

Hence, using $m = O(t^{r+1} \log n)$, the event holds for at least one row for every set of $t + r$ indices with high probability. Therefore, with high probability, the family of sets $\mathcal{F} = \{\mathcal{H}_1, \mathcal{H}_2, \ldots, \mathcal{H}_n\}$ corresponding to the rows of $\boldsymbol{G}$ is a $(r,t) - \mathsf{CFF}$. $\qquad\square$

# B Two-stage Approximate Recovery

In this section, we prove the helper Lemmas 10 and 11 to compete the proof of Theorem 1 and also present the proof of Theorem 2. The two stage approximate recovery algorithm, as the name suggests, proceeds in two sequential steps. In the first stage, we recover the support of all the $\ell$ unknown vectors (presented in Algorithm 2 in Section 5). In the second stage, we use these deduced supports to approximately recover the unknown vectors (Algorithm 5 described in Section B.2).

## B.1 Support recovery (Missing proofs from Section 5)

**Compute $|\mathcal{S}(i)|$ using Algorithm 3.** First, we show how to compute $|\mathcal{S}(i)|$ for every index $i \in [n]$. Let $\mathcal{F} = \{\mathcal{H}_1, \mathcal{H}_2, \ldots, \mathcal{H}_n\}$ be a $(d, \ell k, 0.5)$-RUFF of size $n$ over alphabet $[m]$. Construct the binary matrix $\boldsymbol{A} \in \{0,1\}^{m \times n}$ from $\mathcal{F}$, as $\boldsymbol{A}_{i,j} = 1$ if and only if $i \in \mathcal{H}_j$. Each column $j \in [n]$ of $\boldsymbol{A}$ is essentially the indicator vector of the set $\mathcal{H}_j$. We use the rows of matrix $\boldsymbol{A}$ as query vectors to compute $|\mathcal{S}(i)|$ for each $i \in [n]$. For each such query vector $\boldsymbol{v}$, we compute the nzcount$(\boldsymbol{v})$ using Algorithm 1 with batchsize $T = O(\ell^2 \log \ell k n)$. The large value of $T$ ensures that the estimated nzcount is correct for all the queries with very high probability.

For every $h \in \{0, \ldots, \ell\}$, let $\boldsymbol{b}^h \in \{0,1\}^m$ be the indicator of the queries that have nzcount at least $h$. We show in Lemma 10 that the set of columns of $\boldsymbol{A}$ that have large intersection with $\boldsymbol{b}^h$, exactly correspond to the indices $i \in [n]$ that satisfy $|\mathcal{S}(i)| \geq h$. This allows us to recover $|\mathcal{S}(i)|$ exactly for each $i \in [n]$.

---

**Algorithm 3** COMPUTE–$|\mathcal{S}(i)|$

**Require:** Construct binary matrix $\boldsymbol{A} \in \{0,1\}^{m \times n}$ from $(d, \ell k, 0.5) -$ RUFF of size $n$ over alphabet $[m]$, with $m = c_1 \ell^2 k^2 \log n$ and $d = c_2 \ell k \log n$.
1: Initialize $\boldsymbol{b}^0, \boldsymbol{b}^1, \boldsymbol{b}^2, \ldots, \boldsymbol{b}^\ell$ to all zero vectors of dimension $m$.
2: Let batchsize $T = 4\ell^2 \log mn$.
3: **for** $i = 1, \ldots, m$ **do**
4:     Set $w := \mathsf{nzcount}(\boldsymbol{A}[i])$ (obtained using Algorithm 1 with batchsize $T$.)
5:     **for** $h = 0, 1, \ldots, w$ **do**
6:         Set $\boldsymbol{b}_i^h = 1$.
7:     **end for**
8: **end for**
9: **for** $h = 0, 1, \ldots, \ell$ **do**
10:     Set $\mathcal{C}_h = \{i \in [n] \mid |\mathsf{supp}(\boldsymbol{b}^h) \cap \mathsf{supp}(\boldsymbol{A}_i)| \geq 0.5d\}$.
11: **end for**
12: **for** $i = 1, 2, \ldots, n$ **do**
13:     Set $|\mathcal{S}(i)| = h$ if $i \in \{\mathcal{C}_h \setminus \mathcal{C}_{h+1}\}$ for some $h \in \{0, 1, \ldots, \ell - 1\}$.
14:     Set $|\mathcal{S}(i)| = \ell$ if $i \in \mathcal{C}_\ell$
15: **end for**

---

*Proof of Lemma 10.* Since $\boldsymbol{A}$ has $m = O(\ell^2 k^2 \log n)$ distinct rows, and each row is queried $T = O(\ell^2 \log(mn))$ times, the total query complexity of Algorithm 3 is $O(\ell^4 k^2 \log(\ell k n) \log n)$.

To prove the correctness, we first see that the nzcount for each query is estimated correctly using Algorithm 1 with overwhelmingly high probability. From Lemma 5 with $T = 4\ell^2 \log(mn)$, it follows that each nzcount is estimated correctly with probability at least $1 - \frac{1}{mn^2}$. Therefore, by taking a union bound over all rows of $\boldsymbol{A}$, we estimate all the counts accurately with probability at least $1 - \frac{1}{n^2}$.

We now show, using the properties of RUFF, that $|\mathsf{supp}(\boldsymbol{b}^h) \cap \mathsf{supp}(\boldsymbol{A}_i)| \geq 0.5d$ if and only if $|\mathcal{S}(i)| \geq h$, for any $0 \leq h \leq \ell$.

Let $i \in [n]$ be an index such that $|\mathcal{S}(i)| \geq h$, i.e., there exist at least $h$ unknown vectors that have a non-zero entry in their $i^{th}$ coordinate. Also, let $U := \cup_{i \in [\ell]} \mathsf{supp}(\boldsymbol{\beta}^i)$ denote the union of supports of all the unknown vectors. Since each unknown vector is $k$-sparse, it follows that $|U| \leq \ell k$. To show that $|\mathsf{supp}(\boldsymbol{b}^h) \cap \mathsf{supp}(\boldsymbol{A}_i)| \geq 0.5d$, consider the set of rows of $\boldsymbol{A}$ indexed by $W := \{\mathsf{supp}(\boldsymbol{A}_i) \setminus \cup_{j \in U \setminus \{i\}} \mathsf{supp}(\boldsymbol{A}_j)\}$. Since $\boldsymbol{A}$ is a $(d, \ell k, 0.5) -$ RUFF, we know that $|W| \geq 0.5d$. We now show that $\boldsymbol{b}_t^h = 1$ for every $t \in W$. This follows from the observation that

for $t \in W$, and each unknown vector $\boldsymbol{\beta} \in \mathcal{S}(i)$, the query $\text{sign}(\langle \boldsymbol{A}[t], \boldsymbol{\beta} \rangle) = \text{sign}(\boldsymbol{\beta}_i) \neq 0$. Since $|\mathcal{S}(i)| \geq h$, we conclude that $\text{nzcount}(\boldsymbol{A}[t]) \geq h$, and therefore, $\boldsymbol{b}_t^h = 1$.

To prove the converse, consider an index $i \in [n]$ such that $|\mathcal{S}(i)| < h$. Using a similar argument as above, we now show that $|\text{supp}(\boldsymbol{b}^h) \cap \text{supp}(\boldsymbol{A}_i)| < 0.5d$. Consider the set of rows of $\boldsymbol{A}$ indexed by $W := \{\text{supp}(\boldsymbol{A}_i) \setminus \cup_{j \in U \setminus \{i\}} \text{supp}(\boldsymbol{A}_j)\}$. Now observe that for each $t \in W$, and any unknown vector $\boldsymbol{\beta} \notin \mathcal{S}(i)$, the query $\text{sign}(\langle \boldsymbol{A}[t], \boldsymbol{\beta} \rangle) = 0$. Therefore $\text{nzcount}(\boldsymbol{A}[t]) \leq |\mathcal{S}(i)| < h$, and $\boldsymbol{b}_t^h = 0$ for all $t \in W$. Since $|W| \geq 0.5d$, it follows that $|\text{supp}(\boldsymbol{b}^h) \cap \text{supp}(\boldsymbol{A}_i)| < 0.5d$.

For any $0 \leq h \leq \ell$, Algorithm 3. therefore correctly identifies the set of indices $i \in [n]$ such that $|\mathcal{S}(i)| \geq h$. In particular, the set $C_h := \{i \in [n] \mid |\mathcal{S}(i)| \geq h\}$. Therefore, the set $\mathcal{C}_h \setminus \mathcal{C}_{h+1}$ is exactly the set of indices $i \in [n]$ such that $|\mathcal{S}(i)| = h$.

$\square$

**Compute $|\mathcal{S}(i) \cup \mathcal{S}(j)|$ using Algorithm 4.** In this section we present an algorithm to compute $|\mathcal{S}(i) \cup \mathcal{S}(j)|$, for every $i, j \in [n]$, using $|\mathcal{S}(i)|$ computed in the previous step. We will need an $\ell k - \text{PUFF}$ for this purpose. Let $\mathcal{F} = \{\mathcal{H}_1, \mathcal{H}_2, \ldots, \mathcal{H}_n\}$ be the required $\ell k - \text{PUFF}$ of size $n$ over alphabet $m' = O(\ell^3 k^3 \log n)$.

Construct a set of $\ell + 1$ matrices $\mathcal{B} = \{\boldsymbol{B}^{(1)}, \ldots, \boldsymbol{B}^{(\ell+1)}\}$ where, each $\boldsymbol{B}^{(w)} \in \mathbb{R}^{m' \times n}, w \in [\ell+1]$, is obtained from the PUFF $\mathcal{F}$ in the following way: For every $(i, j) \in [m'] \times [n]$, set $\boldsymbol{B}_{i,j}^{(w)}$ to be a random number sampled uniformly from $[0, 1]$ if $i \in H_j$, and $0$ otherwise. We remark that the choice of uniform distribution in $[0, 1]$ is arbitrary, and any continuous distribution works.

Since every $\boldsymbol{B}^{(w)}$ is generated identically, they have the exact same support, though the non-zero entries are different. Also, by definition, the support of the columns of every $\boldsymbol{B}^{(w)}$ corresponds to the sets in $\mathcal{F}$.

Let $U := \cup_{i \in [\ell]} \text{supp}(\boldsymbol{\beta}^i)$ denote the union of supports of all the unknown vectors. Since each unknown vector is $k$-sparse, it follows that $|U| \leq \ell k$. From the properties of $\ell k - \text{PUFF}$, we know that for any pair of indices $(i, j) \in U \times U$, the set $(\mathcal{H}_i \cap \mathcal{H}_j) \setminus \bigcup_{q \in U \setminus \{i,j\}} \mathcal{H}_q$ is non-empty. This implies that for every $w \in [\ell + 1]$, there exists at least one row of $\boldsymbol{B}^{(w)}$ that has a non-zero entry in the $i^{th}$ and $j^{th}$ index, and $0$ in all other indices $p \in U \setminus \{i, j\}$. In Algorithm 4 we use these rows as queries to estimate their nzcount. In Lemma 11, we show that this quantity is exactly $|S(i) \cup S(j)|$ for that particular pair $(i, j) \in U \times U$.

*Proof of Lemma 11.* Computing each count requires $O(T\ell)$ queries. Therefore, the total number of oracle queries made by Algorithm 4 is at most $O(m'T\ell) = O(\ell^6 k^3 \log(\ell k n) \log n)$ for $m' = O(\ell^3 k^3 \log n)$ and $T = 10\ell^2 \log(nm')$. Also, observe that each nzcount is estimated correctly with probability at least $1 - O(1/\ell m' n^2)$. Therefore from union bound it follows that all the $(\ell + 1)m'$ estimations of nzcount are correct with probability at least $1 - O(1/n^2)$.

Recall that the set $U$ denotes the union of supports of all the unknown vectors. This set is equivalent to $\{i \in [n] \mid |\mathcal{S}(i)| > 0\}$. First, note that if $|\mathcal{S}(i)| = 0$, there are no unknown vectors supported on the $i^{th}$ index. Therefore, $|\mathcal{S}(i) \cup \mathcal{S}(j)| = |\mathcal{S}(j)|$. Also, if $i = j$, then the computation of $|\mathcal{S}(i) \cup \mathcal{S}(j)|$ is trivial.

We now focus on the only non-trivial case when $(i, j) \in U \times U$ and $i \neq j$. Since for every $w \in [\ell + 1]$, the support of the columns of $\boldsymbol{B}^{(w)}$ are the indicators of sets in $\mathcal{F}$, the PUFF property implies that there exists at least one row (say, with index $p \in [m']$) of every $\boldsymbol{B}^{(w)}$ which has a non-zero entry in the $i^{th}$ and $j^{th}$ index, and $0$ in all other indices $q \in U \setminus \{i, j\}$, i.e.,

$$\boldsymbol{B}_{p,i}^{(w)} \neq 0, \boldsymbol{B}_{p,j}^{(w)} \neq 0, \text{ and } \boldsymbol{B}_{p,q}^{(w)} = 0 \text{ for all } q \in U \setminus \{i, j\}.$$

To prove the correctness of the algorithm, we need to show the following:

$$|\mathcal{S}(i) \cup \mathcal{S}(j)| = \max_{w \in [\ell+1]} \{\text{nzcount}(\boldsymbol{B}^{(w)}[p])\}$$

First observe that using the row $\boldsymbol{B}^{(w)}[p]$ as query will produce non-zero value for only those unknown vectors $\beta \in \mathcal{S}(i) \cup \mathcal{S}(j)$. This establishes the fact that $|\mathcal{S}(i) \cup \mathcal{S}(j)| \geq \text{nzcount}(\boldsymbol{B}^{(w)}[p])$.

---

**Algorithm 4** RECOVER$-|\mathcal{S}(i) \cup \mathcal{S}(j)|$

---

**Require:** $|\mathcal{S}(i)|$ for every $i \in [n]$.
**Require:** For every $w \in [\ell+1]$, construct $\boldsymbol{B}^{(w)} \in \mathbb{R}^{m' \times n}$ from $\ell k - \text{PUFF}$ of size $n$ over alphabet
$\quad m' = c_3 \ell^3 k^3 \log n$.
 1: Let $U := \{i \in [n] \mid |\mathcal{S}(i)| > 0\}$
 2: Let batchsize $T = 10\ell^2 \log(nm')$
 3: **for** every $p \in [m']$ **do**
 4: $\quad$ Let $\text{count}(p) := \max_{w \in [\ell+1]} \{\text{nzcount}(\boldsymbol{B}^{(w)}[p])\}$
 $\quad\quad$ (obtained using Algorithm 1 with batchsize $T$).
 5: **end for**
 6: **for** every pair $(i,j) \in [n] \times [n]$ **do**
 7: $\quad$ **if** $i == j$ **then**
 8: $\quad\quad$ Set $|\mathcal{S}(i) \cup \mathcal{S}(j)| = |\mathcal{S}(i)|$
 9: $\quad$ **else if** $i \notin U$ **then**
10: $\quad\quad$ Set $|\mathcal{S}(i) \cup \mathcal{S}(j)| = |\mathcal{S}(j)|$
11: $\quad$ **else if** $j \notin U$ **then**
12: $\quad\quad$ Set $|\mathcal{S}(i) \cup \mathcal{S}(j)| = |\mathcal{S}(i)|$
13: $\quad$ **else**
14: $\quad\quad$ Let $p \in [m']$ such that $\boldsymbol{B}_{p,i}^{(1)} \neq 0$, $\boldsymbol{B}_{p,j}^{(1)} \neq 0$, and $\boldsymbol{B}_{p,q}^{(1)} = 0$ for all $q \in U \setminus \{i,j\}$.
15: $\quad\quad$ Set $|\mathcal{S}(i) \cup \mathcal{S}(j)| = \text{count}(p)$.
16: $\quad$ **end if**
17: **end for**

---

To show the other side of the inequality, consider the set of $(\ell + 1)$ 2-dimensional vectors obtained by the restriction of rows $\boldsymbol{B}^{(w)}[p]$ to the coordinates $(i, j)$,

$$\{(\boldsymbol{B}_{p,i}^{(w)}, \boldsymbol{B}_{p,j}^{(w)}) \mid w \in [\ell+1]\}.$$

Since these entries are picked uniformly at random from $[0, 1]$, they are pairwise linearly independent. Therefore, each $\beta \in \mathcal{S}(i) \cup \mathcal{S}(j)$ can have $\text{sign}(\langle \boldsymbol{B}^{(w)}[p], \beta \rangle) = 0$ for at most 1 of the $w$ queries. So by pigeonhole principle, at least one of the query vectors $\boldsymbol{B}^{(w)}[p]$ will have $\text{sign}(\langle \boldsymbol{B}^{(w)}[p], \beta \rangle) \neq 0$ for all $\beta \in \mathcal{S}(i) \cup \mathcal{S}(j)$. Hence, $|\mathcal{S}(i) \cup \mathcal{S}(j)| \leq \max_w \{\text{nzcount}(\boldsymbol{B}^{(w)}[p])\}$.
$\qquad\qquad\qquad\qquad\qquad\qquad\qquad\qquad\qquad\qquad\qquad\qquad\qquad\qquad\qquad\qquad\square$

## B.2 Approximate Recovery

Once we have the obtained the support of all unknown vectors, the task of approximate recovery can be achieved using a set of *Gaussian queries*. Recall from Definition 4, a Gaussian query refers to an oracle query with vector $\boldsymbol{v} = (\boldsymbol{v}_1, \ldots, \boldsymbol{v}_n) \in \mathbb{R}^n$ where each $\boldsymbol{v}_i$ is sampled independently from the standard Normal distribution, $\boldsymbol{v}_i \sim \mathcal{N}(0, 1)$. The use of Gaussian queries in the context of 1-bit compressed sensing ($\ell = 1$) was studied by [19].

**Lemma 12** ([19]). *For any $\epsilon > 0$, there exists an $\epsilon$-recovery algorithm to efficiently recover an unknown vector in $\mathbb{R}^n$ using $O\left(\frac{n}{\epsilon} \log \frac{n}{\epsilon}\right)$ Gaussian queries.*

In the current query model however, the approximate recovery is a bit intricate since we do not possess the knowledge of the particular unknown vector that was sampled by the oracle. To circumvent this problem, we will leverage the special support structure of the unknown vectors. From Assumption 1, we know that every unknown vector $\boldsymbol{\beta}^t, t \in [\ell]$, has at least one coordinate which is not contained in the support of the other unknown vectors. We will denote the first such coordinate by $\text{rep}(\boldsymbol{\beta}^t)$. Define,

$$\text{rep}(\boldsymbol{\beta}^t) := \min_p \{ \quad p \in \text{supp}(\boldsymbol{\beta}^t) \setminus \bigcup_{q \in [\ell] \setminus \{t\}} \text{supp}(\boldsymbol{\beta}^q) \} \in [n].$$

For $\epsilon$-recovery of a fixed unknown vector $\boldsymbol{\beta}^t$, we will use the set of representative coordinates $\{\text{rep}(\boldsymbol{\beta}^{t'})\}_{t' \neq t}$, to correctly identify its responses with respect to a set of Gaussian queries. In order to achieve this, we first have to recover the sign of $\boldsymbol{\beta}_{\text{rep}(\boldsymbol{\beta}^t)}^t$ for every $t \in [\ell]$, using an RUFF, which is described in Algorithm 6.

**Lemma 13.** *Algorithm 6 recovers* $\mathsf{sign}(\beta^t_{\mathsf{rep}(\beta^t)})$ *for all* $t \in [\ell]$.

With the knowledge of all the supports, and the sign of every representative coordinate, we are now ready to prove Theorem 2. The details are presented in the Algorithm 5.

---

**Algorithm 5** $\epsilon$-RECOVERY, TWO STAGE

---

**Require:** Query access to oracle $\mathcal{O}$.
**Require:** Assumption 1 to be true.
1: Estimate $\mathsf{supp}(\beta^t)$ for all $t \in [\ell]$ using Algorithm 2.
2: Estimate $\mathsf{sign}(\beta^t_{\mathsf{rep}(\beta^t)})$ for all $t \in [\ell]$ using Algorithm 6.
3: Let $\mathsf{Inf}$ be a large positive number.
4: Let batchsize $T = 4\ell^2 \log(nk/\epsilon)$.
5: **for** $t = 1, \ldots, \ell$ **do**
6:    **for** $i = 1, \ldots, \tilde{O}(k/\epsilon)$ **do**
7:       Define $v^t_j := \begin{cases} \mathsf{Inf} & \text{if } j = \mathsf{rep}(\beta^{t'}), \text{for some } t' \neq t \\ \mathcal{N}(0,1) & \text{otherwise} \end{cases}$
8:       Obtain $\mathsf{poscount}(v^t)$ using Algorithm 1 with batchsize $T$.
9:       Let $p_t := |\{t' \neq t \mid \mathsf{sign}(\beta^{t'}_{\mathsf{rep}(\beta^{t'})}) = +1\}|$
10:      **if** $\mathsf{poscount}(v^t) \neq p_t$ **then**
11:         Set $y^t_i = +1$.
12:      **else**
13:         Set $y^t_i = -1$.
14:      **end if**
15:    **end for**
16:    From $\{y^t_1, y^t_2, \ldots, y^t_{\tilde{O}(k/\epsilon)}\}$, and $\mathsf{supp}(\beta^t)$ recover $\hat{\beta}^t$ by using Lemma 12.
17: **end for**
18: Return $\{\hat{\beta}^t, t \in [\ell]\}$.

---

*Proof of Theorem 2.* For the $\epsilon$-recovery of a fixed unknown vector $\beta^t, t \in [\ell]$, we will generate its correct response with respect to a set of $\tilde{O}(k/\epsilon)$ Gaussian queries using *modified* Gaussian queries. A modified Gaussian query $v^t$ for the $t$-th unknown vector, is a Gaussian query with a large positive entry in the coordinates indexed by $\mathsf{rep}(\beta^{t'})$, for every $t' \neq t$.

Consider a fixed unknown vector $\beta^t$. Let $v \in \mathbb{R}^n$ be a Gaussian query, i.e., every entry of $v$ is sampled independently from $\mathcal{N}(0,1)$. Algorithm 5 constructs a modified Gaussian query $v^t$ from $v$ as follows:

$$v^t_j = \begin{cases} \mathsf{Inf} & \text{if} \quad j = \mathsf{rep}(\beta^{t'}) \quad \text{for some } t' \neq t \\ v_j & \text{otherwise} \end{cases}.$$

From construction, we know that $v^t_j = v_j$ for all $j \in \mathsf{supp}(\beta^t)$. Therefore,

$$\langle v^t, \beta^t \rangle = \langle v, \beta^t \rangle \qquad \text{and therefore} \qquad \mathsf{sign}(\langle v^t, \beta^t \rangle) = \mathsf{sign}(\langle v, \beta^t \rangle).$$

On the other hand, if $\mathsf{Inf}$ is chosen to be large enough,

$$\mathsf{sign}(\langle v^t, \beta^{t'} \rangle) = \mathsf{sign}(\beta^{t'}_{\mathsf{rep}(\beta^{t'})}) \qquad \forall t' \neq t,$$

since $\mathsf{Inf} \cdot \beta^{t'}_{\mathsf{rep}(\beta^{t'})}$ dominates the sign of the inner product. Note that in order to obtain an upper bound on the value of $\mathsf{Inf}$, we have to assume that the non-zero entries of every unknown vector have some non-negligible magnitude (at least $1/\mathrm{poly}(n)$).

Note that the $\mathsf{sign}(\beta^{t'}_{\mathsf{rep}(\beta^{t'})})$ was already computed using Algorithm 6, and therefore, the response of the modified Gaussian query with each $\beta^{t'}, t' \neq t$ is known. Now if $\mathsf{poscount}(v^t)$ is different from the number of positive instances of $\mathsf{sign}(\beta^{t'}_{\mathsf{rep}(\beta^{t'})}), t' \neq t$, then it follows that $\mathsf{sign}(\langle v^t, \beta^t \rangle) = +1$. From this we can successfully obtain the response of $\beta^t$ corresponding to a Gaussian query $v$.

Algorithm 5 simulates $O(k/\epsilon \cdot \log(k/\epsilon))$ Gaussian queries for every $\boldsymbol{\beta}^t, t \in [\ell]$ using the modified Gaussian queries $\boldsymbol{v}^t$. Approximate recovery is then possible using Lemma 12 (restricted to the $k$-non zero coordinates in the $\mathsf{supp}(\boldsymbol{\beta}^t)$).

We now argue about the query complexity and the success probability of Algorithm 5.

For every unknown vector $\boldsymbol{\beta}^t, t \in [\ell]$, we simulate $O(k/\epsilon \cdot \log(k/\epsilon))$ Gaussian queries. Simulating each Gaussian query involves $T = O(\ell^2 \log(nk/\epsilon))$ oracle queries to estimate the poscount. Note that Algorithm 6 can be run simultaneously with Algorithm 3 since they use the same set of queries. The sign recovery algorithm, therefore, does not increase the query complexity of approximate recovery. The total query complexity of Algorithm 5 after the support recovery procedure is at most $O\left((\ell^3 k/\epsilon) \log(nk/\epsilon) \log(k/\epsilon)\right)$.

From Lemma 5, each poscount is correct with probability at least $1 - O(\epsilon/(n^2 k^2))$ and therefore by a union bound over all the $O(\ell k/\epsilon \cdot log(k/\epsilon))$ poscount estimates, the algorithm succeeds with probability at least $1 - O(1/n)$. □

*Proof of Lemma 13.* Consider the $(d, \ell k, 0.5) - \mathsf{RUFF}$, $\mathcal{F} = \{\mathcal{H}_1, \mathcal{H}_2, \ldots, \mathcal{H}_n\}$, of size $n$ over alphabet $m = O(\ell^2 k^2 \log n)$ used in Algorithm 3. Let $\boldsymbol{A} \in \{0, 1\}^{m \times n}$ be the binary matrix constructed from the RUFF in a similar manner, i.e., $\boldsymbol{A}_{i,j} = 1$ if and only if $i \in \mathcal{H}_j$. From the properties of RUFF, we know that for every $t \in [\ell]$, there exists a row (indexed by $i \in [m]$) of $\boldsymbol{A}$ such that $\boldsymbol{A}_{i,u(\boldsymbol{\beta}^t)} \neq 0$, and $\boldsymbol{A}_{i,j} = 0$ for all $j \in U \setminus \{u(\boldsymbol{\beta}^t)\}$, where, $U = \cup_{i \in [\ell]} \mathsf{supp}(\boldsymbol{\beta}^i)$. Therefore, the query with $\boldsymbol{A}[i]$ yields non-zero sign with only $\boldsymbol{\beta}^t$. Since,

$$\mathsf{sign}(\langle \boldsymbol{A}[i], \boldsymbol{\beta}^t \rangle) = \mathsf{sign}(\langle \boldsymbol{e}_{u(\boldsymbol{\beta}^t)}, \boldsymbol{\beta}^t \rangle) = \mathsf{sign}(\boldsymbol{\beta}^t_{u(\boldsymbol{\beta}^t)})$$

$\mathsf{sign}(\boldsymbol{\beta}^t_{u(\boldsymbol{\beta}^t)})$ can be deduced.

---

**Algorithm 6** COMPUTE–$\mathsf{sign}(\boldsymbol{\beta}^t_{\mathsf{rep}(\boldsymbol{\beta}^t)})$

---

**Require:** Binary matrix $\boldsymbol{A} \in \{0, 1\}^{m \times n}$ from $(d, \ell k, 0.5) - \mathsf{RUFF}$ of size $n$ over alphabet $[m]$, with $m = O(\ell^2 k^2 \log n)$ and $d = O(\ell k \log n)$.
**Require:** $\mathsf{rep}(\boldsymbol{\beta}^t) \in [n]$ for all $t \in [\ell]$.
 1: Let batchsize $T = 4\ell^2 \log mn$.
 2: Let $U := \cup_{i \in [\ell]} \mathsf{supp}(\boldsymbol{\beta}^i)$.
 3: **for** $t = 1, \ldots, \ell$ **do**
 4:     Let $i \in \{\mathsf{supp}(\boldsymbol{A}_{\mathsf{rep}(\boldsymbol{\beta}^t)}) \setminus \cup_{j \in U \setminus \{\mathsf{rep}(\boldsymbol{\beta}^t)\}} \mathsf{supp}(\boldsymbol{A}_j)\}$
 5:     **if** $\mathsf{poscount}(\boldsymbol{A}[i]) > 0$ (obtained using Algorithm 1 with batchsize $T$.) **then**
 6:         $\mathsf{sign}(\boldsymbol{\beta}^t_{\mathsf{rep}(\boldsymbol{\beta}^t)}) = +1$.
 7:     **else**
 8:         $\mathsf{sign}(\boldsymbol{\beta}^t_{\mathsf{rep}(\boldsymbol{\beta}^t)}) = -1$.
 9:     **end if**
10: **end for**

---

□

## C  Single stage process for $\epsilon$-recovery

The approximate recovery procedure (Algorithm 5), described in Section B.2, crucially utilizes the support information of every unknown vector to design its queries. This requirement forces the algorithm to proceed in two sequential stages.

In particular, Algorithm 5, with the knowledge of the support and the representative coordinates of all the unknown vectors, designed modified Gaussian queries that in turn simulated Gaussian queries for a fixed unknown vector. In this section, we achieve this by using the rows of a matrix obtained from an $(\ell, \ell k) - \mathsf{CFF}$. The property of the CFF allows us to simulate enough Gaussian queries for every unknown vector without the knowledge of their supports. This observation gives us a completely non-adaptive algorithm for approximate recovery of all the unknown vectors.

Consider a matrix $\boldsymbol{A}$ of dimension $m \times n$ constructed from an $(\ell, \ell k)-\mathsf{CFF}$, $\mathcal{F} = \{\mathcal{H}_1, \mathcal{H}_2, \ldots, \mathcal{H}_n\}$ of size $n$ over alphabet $m$, as follows:

$$\boldsymbol{A}_{i,j} = \begin{cases} \mathsf{Inf} & \text{if } i \in \mathcal{H}_j \\ v \sim \mathcal{N}(0, 1) & \text{otherwise} \end{cases}.$$

In Lemma 14, we show that for every unknown vector $\boldsymbol{\beta}^t$, there exists a row of $\boldsymbol{A}$ that simulates the Gaussian query for it. Therefore, using $\tilde{O}(k/\epsilon)$ independent blocks of such queries will ensure sufficient Gaussian queries for every unknown vector which then allows us to approximately recover these vectors.

Recall the definition of a representative coordinate of an unknown vector $\boldsymbol{\beta}^t$,

$$\mathsf{rep}(\boldsymbol{\beta}^t) := \min_p\{ \quad p \in \mathsf{supp}(\boldsymbol{\beta}^t) \setminus \bigcup_{q \in [\ell] \setminus \{t\}} \mathsf{supp}(\boldsymbol{\beta}^q)\} \in [n].$$

**Lemma 14.** *For every $t \in [\ell]$, there exists at least one row $\boldsymbol{v}^t$ in $\boldsymbol{A}$ that simulates a Gaussian query for $\boldsymbol{\beta}^t$, and $\mathsf{sign}(\langle \boldsymbol{v}^t, \boldsymbol{\beta}^{t'} \rangle) = \mathsf{sign}(\boldsymbol{\beta}^{t'}_{\mathsf{rep}(\boldsymbol{\beta}^{t'})})$ for all $t' \neq t$.*

*Proof of Lemma 14.* For any fixed $t \in [\ell]$, consider the set of indices

$$\mathcal{X} = \{\mathsf{rep}(\boldsymbol{\beta}^{t'}) \mid t' \in [\ell] \setminus \{t\}\}.$$

Recall that from the property of $(\ell, \ell k) - \mathsf{CFF}$, we must have

$$\bigcap_{j \in \mathcal{X}} \mathsf{supp}(\boldsymbol{A}_j) \not\subseteq \bigcup_{j \in \cup_{q \in [\ell]} \mathsf{supp}(\boldsymbol{\beta}^q) \setminus \mathcal{X}} \mathsf{supp}(\boldsymbol{A}_j).$$

Therefore, there must exist at least one row $\boldsymbol{v}^t$ in $\boldsymbol{A}$ which has a large positive entry, $\mathsf{Inf}$, in all the coordinates indexed by $\mathcal{X}$. Moreover, $\boldsymbol{v}^t$ has a random Gaussian entry in all the other coordinates indexed by the union of support of all unknown vectors. Since $\boldsymbol{\beta}^t$ is 0 for all coordinates in $\mathcal{X}$, the query $\mathsf{sign}(\langle \boldsymbol{v}^t, \boldsymbol{\beta}^t \rangle)$ simulates a Gaussian query. Also,

$$\mathsf{sign}(\langle \boldsymbol{v}, \boldsymbol{\beta}^{t'} \rangle) = \mathsf{sign}(\mathsf{rep}(\boldsymbol{\beta}^{t'})) \qquad \forall t' \neq t$$

since $\mathsf{Inf} \times \boldsymbol{\beta}^{t'}_{\mathsf{rep}(\boldsymbol{\beta}^{t'})}$ dominates the inner product. $\qquad\qquad\square$

We are now ready to present the completely non-adaptive algorithm for the approximate recovery of all the unknown vectors.

*Proof of Theorem 3.* The proof of Theorem 3 follows from the guarantees of Algorithm 7. The query vectors of Algorithm 7 can be represented by the rows of the following matrix:

$$\boldsymbol{R} = \begin{bmatrix} \boldsymbol{A} \\ \tilde{\boldsymbol{A}} + \boldsymbol{B}^{(1)} \\ \tilde{\boldsymbol{A}} + \boldsymbol{B}^{(2)} \\ \vdots \\ \tilde{\boldsymbol{A}} + \boldsymbol{B}^{(\mathsf{D})} \end{bmatrix}$$

where, $\mathsf{D} = O(k/\epsilon \cdot \log k/\epsilon)$ and $\boldsymbol{A}$ is the matrix obtained from the $(d, \ell k, 0.5) - \mathsf{RUFF}$ required by Algorithm 2 and Algorithm 6. The matrix $\tilde{\boldsymbol{A}}$ is obtained from an $(\ell, \ell k) - \mathsf{CFF}$, $\mathcal{F} = \{\mathcal{H}_1, \mathcal{H}_2, \ldots, \mathcal{H}_n\}$ by setting $\tilde{\boldsymbol{A}}_{i,j} = \mathsf{Inf}$ if $i \in \mathcal{H}_j$ and 0 otherwise, and each matrix $\boldsymbol{B}^{(w)}$ for $w \in [\mathsf{D}]$ is a Gaussian matrix with every entry $\boldsymbol{B}^{(w)}_{i,j}$ drawn uniformly at random from standard Normal distribution.

Algorithm 7 decides all its query vectors at the start and hence is completely non-adaptive. It first invokes Algorithm 2 and Algorithm 6 to recover the support and the sign of the representative coordinate of every unknown vector $\boldsymbol{\beta}^t$. Now using the queries from the rows of the matrix $\boldsymbol{R}$, the algorithm generates at least $\mathsf{D} = \tilde{O}(k/\epsilon)$ Gaussian queries for each unknown vector.

**Algorithm 7** $\epsilon$-RECOVERY, SINGLE STAGE

---

**Require:** Assumption 1 to be true.
**Require:** Binary matrix $\tilde{\boldsymbol{A}} \in \{0,1\}^{m \times n}$ from $(\ell, \ell k) - \mathsf{CFF}$ of size $n$ over alphabet $m = O((\ell k)^{\ell+1} \log n)$.
1: Estimate $\mathsf{supp}(\boldsymbol{\beta}^t)$ and $\mathsf{sign}(\boldsymbol{\beta}^t_{\mathsf{rep}(\boldsymbol{\beta}^t)})$ for all $t \in [\ell]$ using Algorithm 2 and Algorithm 6 respectively.
2: Set $\mathsf{Inf}$ to be a large positive number.
3: Set $\mathsf{D} = O(k/\epsilon \cdot \log(k/\epsilon))$.
4: Set batchsize $T = 4\ell^2 \log(mnk/\epsilon)$.
5: **for** $i = 1, \ldots, m$ **do**
6:     **for** $w = 1, 2, \ldots, \mathsf{D}$ **do**
7:         Construct query vector $\boldsymbol{v}$, where $\boldsymbol{v}_j = \begin{cases} \mathsf{Inf} & \text{if } \tilde{\boldsymbol{A}}_{i,j} = 1 \\ \mathcal{N}(0,1) & \text{otherwise} \end{cases}$.
8:         $\mathsf{Query}\Big(\boldsymbol{v}, T\Big)$ and set $\boldsymbol{P}_{i,w} = \mathsf{poscount}(\boldsymbol{v})$.
9:     **end for**
10: **end for**
11: **for** $t = 1, \ldots, \ell$ **do**
12:     Let $\mathcal{X} := \{\mathsf{rep}(\boldsymbol{\beta}^{t'}) \mid t' \in [\ell] \setminus t\}$ and $U := \cup_q \mathsf{supp}(\boldsymbol{\beta}^q)$
13:     Let $i \in \{\cap_{j \in \mathcal{X}} \mathsf{supp}(\tilde{\boldsymbol{A}}_j) \setminus \bigcup_{j \in U \setminus \mathcal{X}} \mathsf{supp}(\tilde{\boldsymbol{A}}_j)\} \subset [m]$.
14:     Let $p := |\{t' \neq t \mid \mathsf{sign}(\boldsymbol{\beta}^t_{\mathsf{rep}(\boldsymbol{\beta}^t)}) = +1\}|$
15:     **for** $w = 1, \ldots, \mathsf{D}$ **do**
16:         **if** $\boldsymbol{P}_{i,w} \neq p$ **then**
17:             Set $y^t_w = +1$
18:         **else**
19:             Set $y^t_w = -1$
20:         **end if**
21:     **end for**
22:     From $\{y^t_w \mid w \in [\mathsf{D}]\}$ and $\mathsf{supp}(\boldsymbol{\beta}^t)$ recover $\hat{\boldsymbol{\beta}}^t$ by using Lemma 12.
23: **end for**
24: Return $\{\hat{\boldsymbol{\beta}}^t \mid t \in [\ell]\}$.

---

It follows from Lemma 14 that each matrix $\tilde{\boldsymbol{A}} + \boldsymbol{B}^{(w)}$, for $w \in [\mathsf{D}]$, contains at least one Gaussian query for every unknown vector. Therefore, in total, $\boldsymbol{R}$ contains at least $D = O(k/\epsilon \cdot \log k/\epsilon)$ Gaussian queries for every unknown vector $\boldsymbol{\beta}^t$. Using the responses of these Gaussian queries, we can then approximately recover every $\boldsymbol{\beta}^t$ using Lemma 12.

The total query complexity is therefore the sum of query complexities of support recovery process (which from Theorem 1 we know to be at most $O(\ell^6 k^3 \log(n) \log(\ell k n))$), and the total number of queries needed to generate $O(k/\epsilon \cdot \log(k/\epsilon))$ Gaussian queries (which is $mTD$) for each unknown vector. Therefore the net query complexity is $O\Big((\ell^{\ell+3} k^{\ell+2}/\epsilon) \log n \log(k/\epsilon) \log(n/\epsilon))\Big)$. Each Algorithm 2, 6 and the Gaussian query generation succeed with probability at least $1 - O(1/n)$, therefore from union bound, Algorithm 7 succeeds with probability at least $1 - O(1/n)$. $\qquad\square$

## D   Relaxing Assumption 1 for $\ell = 2$

In this section, we will circumvent the necessity for Assumption 1 when there are only two unknown vectors - $\{\boldsymbol{\beta}^1, \boldsymbol{\beta}^2\}$. We present a two-stage algorithm to approximately recover both the unknown vectors. In the first stage, the algorithm recovers the support of both the vectors, and then using the support information it approximately recovers the two vectors.

We would like to mention that if $\mathsf{supp}(\boldsymbol{\beta}^1) \neq \mathsf{supp}(\boldsymbol{\beta}^2)$, we do not need any further assumptions on the unknown vectors for their approximate recovery. However, if the two vectors have the exact same support, then we need to impose some mild assumptions in order to approximately recover the vectors.

## D.1 Support Recovery

In this section, we show that supports of both the unknown vectors can be inferred directly from $\{|\mathcal{S}(i)|\}_{i \in [n]}$ and $\{|\mathcal{S}(i) \cap \mathcal{S}(j)|\}_{i,j \in [n]}$. These quantities were computed using Algorithm 3 and using Algorithm 4 respectively. Moreover, the guarantees of both these algorithms (shown in Lemma 10, and Lemma 11) do not require the unknown vectors to satisfy any special assumption.

**Lemma 15.** *There exists an algorithm to recover the support of any two $k$-sparse unknown vectors using $O(k^3 \log^2 n)$ oracle queries with probability at least $1 - O(1/n^2)$.*

*Proof of Lemma 15.* Consider Algorithm 8. The query complexity and success guarantees both follow from Lemma 10 and Lemma 11. We now prove the correctness of Algorithm 8.

---

**Algorithm 8** RECOVER–SUPPORT $\ell = 2$

---

**Require:** Access to oracle $\mathcal{O}$
1:  Estimate $|S(i)|$ for every $i \in [n]$ using Algorithm 3.
2:  Estimate $|S(i) \cap S(j)|$ for every $i, j \in [n]$ using Algorithm 4.
3:  **if** $|S(i)| \in \{0, 2\}$ for all $i \in [n]$ **then**
4:      $\mathsf{supp}(\boldsymbol{\beta}^1) = \mathsf{supp}(\boldsymbol{\beta}^2) = \{i \in [n] | |\mathcal{S}(i)| \neq 0\}$.
5:  **else**
6:      Let $i_0 = \min\{i | |S(i)| = 1\}$, and let $i_0 \in \mathsf{supp}(\boldsymbol{\beta}^1)$
7:      **for** $j \in [n] \setminus \{i_0\}$ **do**
8:          **if** $|S(j)| = 2$ **then**
9:              Add $j$ to $\mathsf{supp}(\boldsymbol{\beta}^1)$, and $\mathsf{supp}(\boldsymbol{\beta}^2)$.
10:         **else if** $|S(j)| = 1$ and $|\mathcal{S}(i_0) \cap \mathcal{S}(j)| = 0$ **then**
11:             Add $j$ to $\mathsf{supp}(\boldsymbol{\beta}^2)$.
12:         **else if** $|S(j)| = 1$ and $|\mathcal{S}(i_0) \cap \mathcal{S}(j)| = 1$ **then**
13:             Add $j$ to $\mathsf{supp}(\boldsymbol{\beta}^1)$.
14:         **end if**
15:     **end for**
16: **end if**

---

**Case 1: ($\mathsf{supp}(\boldsymbol{\beta}^1) \neq \mathsf{supp}(\boldsymbol{\beta}^2)$).** First note that the set of coordinates, $i \in [n]$ with $|\mathcal{S}(i)| = 2$ belong to the support of both the unknown vectors. For the remaining indices in $T := \{i \in [n] | |S(i)| = 1\}$, we use the following approach to decide the unknown vector whose support they belongs to.

If $|T| = 1$, then without loss of generality we can assume $i \in \mathsf{supp}(\boldsymbol{\beta}^1)$. Else if $|T| > 1$, we set the smallest index $i_0 \in T$ to be in $\mathsf{supp}(\boldsymbol{\beta}^1)$. We then use this index as a pivot to figure out all the other indices $j \in T \cap \mathsf{supp}(\boldsymbol{\beta}^1)$. If both $i_0$, and $j$ lie in $\mathsf{supp}(\boldsymbol{\beta}^1)$, then $|\mathcal{S}(i_0) \cap \mathcal{S}(j)| = 1$, otherwise $|\mathcal{S}(i_0) \cap \mathcal{S}(j)| = 0$. So, using Algorithm 8, we can identify the supports of both the unknown vectors.

**Case 2: ($\mathsf{supp}(\boldsymbol{\beta}^1) = \mathsf{supp}(\boldsymbol{\beta}^2)$).** In this case, we observe that $|\mathcal{S}(i)| \in \{2, 0\}$ for all $i \in [n]$. Therefore, both the unknown vectors have the exact same support, and nothing further needs to be done since $\mathsf{supp}(\boldsymbol{\beta}^1) = \mathsf{supp}(\boldsymbol{\beta}^2) = \{i \in [n] | |\mathcal{S}(i)| \neq 0\}$. $\qquad \square$

## D.2 Approximate Recovery

In this section, we present the approximate recovery algorithm. The queries are designed based on the supports of the two vectors.

We split the analysis in two parts. First, we consider the case when the two vectors have different supports, i.e. $\mathsf{supp}(\boldsymbol{\beta}^1) \neq \mathsf{supp}(\boldsymbol{\beta}^2)$. In this case, we use Lemma 16 to approximately recover the two vectors.

**Lemma 16.** *If $\mathsf{supp}(\boldsymbol{\beta}^1) \neq \mathsf{supp}(\boldsymbol{\beta}^2)$, then there exists an algorithm for $\epsilon$-approximate recovery of any two $k$-sparse unknown vectors using $O\left(\frac{k}{\epsilon} \cdot \log(\frac{nk}{\epsilon})\right)$ oracle queries with probability at least $1 - O(1/n)$.*

When the two vectors have the exact same support, we use a set of sub-Gaussian queries to recover the two vectors. This is slightly tricky, and our algorithms succeeds under some mild assumption on the two unknown vectors (Assumption 2).

**Lemma 17.** *If* $\mathsf{supp}(\boldsymbol{\beta}^1) = \mathsf{supp}(\boldsymbol{\beta}^2)$, *then there exists an algorithm for $\epsilon$-approximate recovery of any two $k$-sparse unknown vectors using $O(\frac{k^2}{\epsilon^4\delta^2}\log^2(\frac{nk}{\delta}))$ oracle queries with probability at least $1 - O(1/n)$.*

---

**Algorithm 9** $\epsilon$-APPROXIMATE-RECOVERY

1: Estimate $\mathsf{supp}(\boldsymbol{\beta}^1), \mathsf{supp}(\boldsymbol{\beta}^2)$ using Algorithm 8.
2: **if** $\mathsf{supp}(\boldsymbol{\beta}^1) \neq \mathsf{supp}(\boldsymbol{\beta}^2)$ **then**
3:     Return $\hat{\boldsymbol{\beta}}^1, \hat{\boldsymbol{\beta}}^2$ using Algorithm 10.
4: **else**
5:     Return $\hat{\boldsymbol{\beta}}^1, \hat{\boldsymbol{\beta}}^2$ using Algorithm 11.
6: **end if**

---

*Proof of Theorem 4.* The guarantees of Algorithm 9 prove Theorem 4. The total query complexity after support recovery is the maximum of the query complexities of Algorithm 10 and Algorithm 11, which is $O(\frac{k^2}{\epsilon\delta^2}\log^2(\frac{nk}{\delta}))$.

Moreover from Lemma 16 and Lemma 17, we know that both these algorithms succeed with a probability at least $1 - O(1/n)$, therefore, Algorithm 9 is also guaranteed to succeed with probability at least $1 - O(1/n)$. □

We now prove Lemma 16 and Lemma 17.

**D.2.1 Case 1:** $\mathsf{supp}(\boldsymbol{\beta}^1) \neq \mathsf{supp}(\boldsymbol{\beta}^2)$**.**

*Proof of Lemma 16.* Consider a coordinate $p \in \mathsf{supp}(\boldsymbol{\beta}^1) \, \Delta \, \mathsf{supp}(\boldsymbol{\beta}^2)$, where $\Delta$ denotes the symmetric difference of the two support sets. Without loss of generality we can assume $p \in \mathsf{supp}(\boldsymbol{\beta}^1)$. We first identify the $\mathsf{sign}(\boldsymbol{\beta}_p^1)$ simply using the query vector $\boldsymbol{e}_p$. For the sake of simplicity let us assume $\mathsf{sign}(\boldsymbol{\beta}_p^1) = +1$.

We use two types of queries to recover the two unknown vectors. The *Type 1* queries are modified Gaussian queries, of the form $\boldsymbol{v} + \mathsf{Inf} \cdot \boldsymbol{e}_p$, where $\boldsymbol{v}$ is a Gaussian query vector. *Type 2* query is the plain Gaussian query $\boldsymbol{v}$.

Since $p \in \mathsf{supp}(\boldsymbol{\beta}^1) \setminus \mathsf{supp}(\boldsymbol{\beta}^2)$, the Type 1 queries will always have a positive response with the unknown vector $\boldsymbol{\beta}^1$. Moreover, they will simulate a Gaussian query with $\boldsymbol{\beta}^2$. Therefore from the responses of the oracle, we can correctly identify the response of $\boldsymbol{\beta}^2$ with a set of $O(k/\epsilon \cdot \log(k/\epsilon))$ Gaussian queries. Now, using Lemma 12, we can approximately recover it.

Now since the response of $\boldsymbol{\beta}^2$ with the Type 1 query $\boldsymbol{v} + \mathsf{Inf} \cdot \boldsymbol{e}_p$ and the corresponding Type 2 query $\boldsymbol{v}$, remains the same, we can also obtain correct responses of $\boldsymbol{\beta}^1$ with a set of $O(k/\epsilon \cdot \log(k/\epsilon))$ Gaussian queries. By invoking Lemma 12 again, we can approximately recover $\boldsymbol{\beta}^1$.

The total query complexity of the algorithm is $O(kT/\epsilon \cdot \log(k/\epsilon)) = O(k/\epsilon \cdot \log(nk/\epsilon) \cdot \log(k/\epsilon))$. Also, from Lemma 5, it follows that each oracle query succeeds with probability at least $1 - O(1/mn)$. Therefore by union bound over all $2m$ queries, the algorithm succeeds with probability at least $1 - O(1/n)$. □

**D.2.2 Case 2:** $\mathsf{supp}(\boldsymbol{\beta}^1) = \mathsf{supp}(\boldsymbol{\beta}^2)$**.**

We now propose an algorithm for approximate recovery of the two unknown vectors when their supports are exactly the same. Until now for $\epsilon$-recovery, we were using a representative coordinate to generate enough responses to Gaussian queries. However, when the supports are exactly the same, the same trick does not work.

**Algorithm 10** $\epsilon$-APPROXIMATE-RECOVERY: CASE 1

---

**Require:** $\mathsf{supp}(\boldsymbol{\beta}^1) \neq \mathsf{supp}(\boldsymbol{\beta}^2)$
1: Set $m = O(k/\epsilon \cdot \log(k/\epsilon))$
2: Set batchsize $T = 10 \log mn$.
3: Let $\mathsf{Inf}$ be a large positive number.
4: Let $p \in \mathsf{supp}(\boldsymbol{\beta}^1) \setminus \mathsf{supp}(\boldsymbol{\beta}^2)$, and $s := \mathsf{sign}(\boldsymbol{\beta}_p^1)$.
5: **for** $i = 1, \ldots, m$ **do**
6:     Construct query vector $\boldsymbol{v}$, where $\boldsymbol{v}_j = \mathcal{N}(0,1)$ for all $j \in [n]$.
7:     Construct query vector $\tilde{\boldsymbol{v}} := \boldsymbol{v} + s \cdot \mathsf{Inf} \cdot \boldsymbol{e}_p$
8:     $\mathsf{Query}\Big(\boldsymbol{v}, T\Big)$, and $\mathsf{Query}\Big(\tilde{\boldsymbol{v}}, T\Big)$.

9:     Set $y^i = \begin{cases} +1 & \text{if } \mathsf{poscount}(\tilde{\boldsymbol{v}}) == 2 \\ -1 & \text{if } \mathsf{negcount}(\tilde{\boldsymbol{v}}) == 1 \\ 0 & \text{otherwise} \end{cases}$

10:    Set $z^i = \begin{cases} +1 & \text{if } y^i = +1 \text{ and } \mathsf{poscount}(\boldsymbol{v}) == 2 \\ -1 & \text{if } y^i = +1 \text{ and } \mathsf{negcount}(\boldsymbol{v}) == 1 \\ +1 & \text{if } y^i = -1 \text{ and } \mathsf{poscount}(\boldsymbol{v}) == 1 \\ -1 & \text{if } y^i = -1 \text{ and } \mathsf{negcount}(\boldsymbol{v}) == 2 \\ +1 & \text{if } y^i = 0 \text{ and } \mathsf{poscount}(\boldsymbol{v}) == 1 \\ -1 & \text{if } y^i = 0 \text{ and } \mathsf{negcount}(\boldsymbol{v}) == 1 \\ 0 & \text{otherwise} \end{cases}$

11: **end for**
12: From $\{y^i \mid i \in [m]\}$ and $\mathsf{supp}(\boldsymbol{\beta}^2)$ recover $\hat{\boldsymbol{\beta}}^2$ by using Lemma 12.
13: From $\{z^i \mid i \in [m]\}$ and $\mathsf{supp}(\boldsymbol{\beta}^1)$ recover $\hat{\boldsymbol{\beta}}^1$ by using Lemma 12.

---

For the approximate recovery in this case, we use sub-Gaussian queries instead of Gaussian queries. In particular, we consider queries whose entries are sampled uniformly from $\{-1, 1\}$. The equivalent of Lemma 12 proved by [2] for sub-Gaussian queries enables us to achieve similar bounds.

**Lemma 18** (Corollary of Theorem 1.1 of [2]). *Let $\boldsymbol{x} \in \mathbb{S}^{n-1}$ be a $k$-sparse unknown vector of unit norm. Let $\boldsymbol{v_1}, \ldots, \boldsymbol{v_m}$ be independent random vectors in $\mathbb{R}^n$ whose coordinates are drawn uniformly from $\{-1, 1\}$. There exists an algorithm that recovers $\hat{\boldsymbol{x}} \in \mathbb{S}^{n-1}$ using the 1-bit sign measurements $\{\mathsf{sign}(\langle \boldsymbol{v_i}, \boldsymbol{x} \rangle)\}_{i \in [m]}$, such that with probability at least $1 - 4e^{-\alpha^2}$ (for any $\alpha > 0$), it satisfies*

$$\|\boldsymbol{x} - \hat{\boldsymbol{x}}\|_2^2 \leq O\left(\|x\|_\infty^{\frac{1}{2}} + \frac{1}{2\sqrt{m}}(\sqrt{k \log(2n/k)} + \alpha)\right).$$

In particular, for $m = O(\frac{k}{\epsilon^4} \log n)$, we get $O(\epsilon + \|x\|_\infty^{\frac{1}{2}})$ - approximate recovery with probability at least $1 - O(1/n)$. Therefore, if the unknown vectors are not *extremely* sparse (Assumption 2), we can get good guarantees on their approximate recovery with sufficient number of sub-Gaussian queries.

The central idea of $\epsilon$-recovery algorithm (Algorithm 11) is therefore to identify the responses of a particular unknown vector $\boldsymbol{\beta}$ with respect to a set of sub-Gaussian queries $\boldsymbol{v} \sim \{-1, 1\}^n$. Then using Lemma 18, we can approximately reconstruct $\boldsymbol{\beta}$.

Let us denote by $\mathsf{response}(\boldsymbol{v})$, the set of distinct responses of the oracle with a query vector $\boldsymbol{v}$. Since there are only two unknown vectors, $|\mathsf{response}(\boldsymbol{v})| \leq 2$. If both unknown vectors have the same response with respect to a given query vector $\boldsymbol{v}$, i.e., $|\mathsf{response}(\boldsymbol{v})| = 1$ then we can trivially identify the correct responses with respect both the unknown vectors by setting $\mathsf{sign}(\langle \boldsymbol{v}, \boldsymbol{\beta}^2 \rangle) = \mathsf{sign}(\langle \boldsymbol{v}, \boldsymbol{\beta}^2 \rangle) = \mathsf{response}(\boldsymbol{v})$.

However if $|\mathsf{response}(\boldsymbol{v})| = 2$, we need to identify the correct response with respect to a fixed unknown vector. This *alignment* constitutes the main technical challenge in approximate recovery. To achieve this, Algorithm 11 fixes a pivot query say $\boldsymbol{v}_0$ with $|\mathsf{response}(\boldsymbol{v}_0)| = 2$, and aligns all the other queries with respect to it by making some additional oracle queries.

Let $W$ denote the set of queries such that $|\mathsf{response}(\boldsymbol{v})| = 2$. Also, for any pair of query vectors, $\boldsymbol{v}_1, \boldsymbol{v}_2 \in W$, we denote by $\mathsf{align}_{\boldsymbol{\beta}}(\boldsymbol{v}_1, \boldsymbol{v}_2)$ to be an ordered tuple of responses with respect to the unknown vector $\boldsymbol{\beta}$.

$$\mathsf{align}_{\boldsymbol{\beta}}(\boldsymbol{v}_1, \boldsymbol{v}_2) = (\mathsf{sign}(\langle \boldsymbol{v}_1, \boldsymbol{\beta} \rangle), \mathsf{sign}(\langle \boldsymbol{v}_2, \boldsymbol{\beta} \rangle)).$$

We fix a pivot query $\boldsymbol{v}_0 \in W$ to be one that satisfies $\mathsf{response}(\boldsymbol{v}_0) = \{-1, 1\}$. We can assume without loss of generality that there always exists one such query, otherwise all queries $\boldsymbol{v} \in W$ have $0 \in \mathsf{response}(\boldsymbol{v})$, and Proposition 19 aligns all such responses using $O(\log n)$ additional oracle queries.

**Proposition 19.** *Suppose for all queries $\boldsymbol{v} \in W$, $0 \in \mathsf{response}(\boldsymbol{v})$. There exists an algorithm that estimates $\mathsf{align}_{\boldsymbol{\beta}^1}(\boldsymbol{v}_0, \boldsymbol{v})$ and $\mathsf{align}_{\boldsymbol{\beta}^2}(\boldsymbol{v}_0, \boldsymbol{v})$ for any $\boldsymbol{v}, \boldsymbol{v}_0 \in W$ using $O(\log n)$ oracle queries with probability at least $1 - O(1/n)$.*

For a fixed pivot query $\boldsymbol{v}_0 \in W$ such that $\mathsf{response}(\boldsymbol{v}_0) = \{-1, 1\}$, Proposition 20 and Proposition 21 compute $\mathsf{align}_{\boldsymbol{\beta}}(\boldsymbol{v}_0, \boldsymbol{v})$ for all queries $\boldsymbol{v} \in W$ such that $0 \in \mathsf{response}(\boldsymbol{v})$ and $0 \notin \mathsf{response}(\boldsymbol{v})$ respectively.

**Proposition 20.** *Let $\boldsymbol{v}_0 \in W$ such that $\mathsf{response}(\boldsymbol{v}_0) = \{-1, 1\}$. For any query vector $\boldsymbol{v} \in W$ such that $0 \in \mathsf{response}(\boldsymbol{v})$, there exists an algorithm that computes $\mathsf{align}_{\boldsymbol{\beta}^1}(\boldsymbol{v}_0, \boldsymbol{v})$ and $\mathsf{align}_{\boldsymbol{\beta}^2}(\boldsymbol{v}_0, \boldsymbol{v})$ using $O(\log n)$ oracle queries with probability at least $1 - O(1/n)$.*

**Proposition 21.** *Let $\delta > 0$, be the largest real number such that $\boldsymbol{\beta}^1, \boldsymbol{\beta}^2 \in \delta \mathbb{Z}^n$. Let $\boldsymbol{v}_0 \in W$ such that $\mathsf{response}(\boldsymbol{v}_0) = \{-1, 1\}$. For any query vector $\boldsymbol{v} \in W$ such that $\mathsf{response}(\boldsymbol{v}) = \{-1, 1\}$, there exists an algorithm that computes $\mathsf{align}_{\boldsymbol{\beta}^1}(\boldsymbol{v}_0, \boldsymbol{v})$ and $\mathsf{align}_{\boldsymbol{\beta}^2}(\boldsymbol{v}_0, \boldsymbol{v})$ using $O(\frac{k}{\delta^2} \log(\frac{nk}{\delta}))$ oracle queries with probability at least $1 - O(1/n)$.*

Using the alignment process and Lemma 18, we can now approximately recover both the unknown vectors.

*Proof of Lemma 17.* Consider Algorithm 11, which basically collects enough responses of an unknown vector for a set of sub-Gaussian queries by aligning all responses.

Without loss of generality, we fix $\boldsymbol{v}_0$ such that $\mathsf{response}(\boldsymbol{v}_0) = \{+1, -1\}$, and also enforce that $\mathsf{sign}(\boldsymbol{v}_0, \boldsymbol{\beta}^1) = +1$. Now, we align all other responses with respect to $\boldsymbol{v}_0$. The proof of Lemma 17 then follows from the guarantees of Lemma 18. For $m = O(\frac{k}{\epsilon^4} \log n)$, along with the assumptions that $\|\boldsymbol{\beta}^1\|_\infty, \|\boldsymbol{\beta}^2\|_\infty = o(1)$, the algorithm approximately recovers $\boldsymbol{\beta}^1, \boldsymbol{\beta}^2$.

The number of queries made by Algorithm 11 is at most $mT$ to generate responses and $O(m \frac{k}{\delta^2} \log(\frac{nk}{\delta}))$ to align all the $m$ responses with respect to a fixed pivot query $\boldsymbol{v}_0$. Therefore the total query complexity of Algorithm 11 is $O(\frac{k^2}{\epsilon^4 \delta^2} \log^2(\frac{nk}{\delta}))$.

All parts of the algorithm succeed with probability at least $1 - O(1/n)$, and therefore the algorithm succeeds with probability at least $1 - O(1/n)$. $\square$

Finally, we prove Proposition 19, Proposition 20 and Proposition 21.

*Proof of Proposition 19.* For the proof of Proposition 19, we simply use the query vector $\boldsymbol{v}_0 + \boldsymbol{v}$ to reveal whether the 0's in the two response sets correspond to the same unknown vector or different ones. The correctness of Algorithm 12 follows from the fact that there will be a 0 in the response set of $\boldsymbol{v}_0 + \boldsymbol{v}$ if and only if both the 0's correspond to the same unknown vector.

To obtain the complete response set for the query $\boldsymbol{v}_0 + \boldsymbol{v}$ with probability at least $1 - 1/n$, Algorithm 12 makes at most $O(\log n)$ queries.

$\square$

*Proof of Proposition 20.* In this case, we observe that the response set corresponding to the query $\mathsf{Inf} \cdot \boldsymbol{v} + \boldsymbol{v}_0$ can reveal the correct alignment. To see this, let the response of $\boldsymbol{v}_0$ and $\boldsymbol{v}$ be $\{+1, -1\}$ and $\{\mathsf{s}, 0\}$ respectively for some $\mathsf{s} \in \{\pm 1\}$. The response set corresponding to $\mathsf{Inf} \cdot \boldsymbol{v} + \boldsymbol{v}_0$ will be the set (or multi-set) of the form $\{\mathsf{s}, \mathsf{t}\}$. Since we know $\mathsf{s} = \mathsf{response}(\boldsymbol{v}) \setminus \{0\}$, we can deduce $\mathsf{t}$ from the $\mathsf{poscount}(\mathsf{Inf} \cdot \boldsymbol{v} + \boldsymbol{v}_0)$, and $\mathsf{negcount}(\mathsf{Inf} \cdot \boldsymbol{v} + \boldsymbol{v}_0)$.

**Algorithm 11** $\epsilon$-APPROXIMATE RECOVERY: CASE 2

**Require:** $\mathrm{supp}(\boldsymbol{\beta}^1) = \mathrm{supp}(\boldsymbol{\beta}^2)$, Assumption 2.
1: Set $m = O(\frac{k}{\epsilon^4}\log(n))$
2: Set batchsize $T = O(\log mn)$
3: **for** $i = 1, \ldots, m$ **do**
4:     Sample query vector $\boldsymbol{v}$ as: $\boldsymbol{v}_j = \begin{cases} +1 & \text{w.p. } 1/2 \\ -1 & \text{w.p. } 1/2 \end{cases}$
5:     Query$(\boldsymbol{v}, T)$, and store response$(\boldsymbol{v})$.
6:     **if** $|\text{response}(\boldsymbol{v})| == 1$ **then**
7:         Set $y^{\boldsymbol{v}} = \text{response}(\boldsymbol{v})$.
8:         Set $z^{\boldsymbol{v}} = \text{response}(\boldsymbol{v})$.
9:     **else**
10:        Add $\boldsymbol{v}$ to $W$.
11:     **end if**
12:     Let $\boldsymbol{v}_0$ be an arbitrary $\boldsymbol{v} \in W$.
13:     **for** every $\boldsymbol{v} \in W$ **do**
14:        Set $(y^{\boldsymbol{v}_0}, y^{\boldsymbol{v}}) = \text{align}_{\boldsymbol{\beta}^1}(\boldsymbol{v}_0, \boldsymbol{v})$.
15:        Set $(z^{\boldsymbol{v}_0}, z^{\boldsymbol{v}}) = \text{align}_{\boldsymbol{\beta}^2}(\boldsymbol{v}_0, \boldsymbol{v})$.
16:     **end for**
17: **end for**
18: Using $\{y^{\boldsymbol{v}}\}_{\boldsymbol{v}}$, estimate $\hat{\boldsymbol{\beta}}^1$.
19: Using $\{z^{\boldsymbol{v}}\}_{\boldsymbol{v}}$, estimate $\hat{\boldsymbol{\beta}}^2$.

---

**Algorithm 12** ALIGN QUERIES, CASE 1

**Require:** $\boldsymbol{v}_0, \boldsymbol{v} \in \{-1, 1\}^n$, $0 \in \text{response}(\boldsymbol{v}_0) \cap \text{response}(\boldsymbol{v})$.
1: Set batchsize $T = O(\log n)$.
2: Query$(\boldsymbol{v}_0 + \boldsymbol{v}, T)$.
3: **if** $0 \in \text{response}(\boldsymbol{v}_0 + \boldsymbol{v})$ **then**
4:     $\text{align}_{\boldsymbol{\beta}^1}(\boldsymbol{v}_0, \boldsymbol{v}) = (0, 0)$
5:     $\text{align}_{\boldsymbol{\beta}^2}(\boldsymbol{v}_0, \boldsymbol{v}) = (\text{response}(\boldsymbol{v}_0) \setminus \{0\}, \text{response}(\boldsymbol{v}) \setminus \{0\})$
6: **else**
7:     $\text{align}_{\boldsymbol{\beta}^1}(\boldsymbol{v}_0, \boldsymbol{v}) = (0, \text{response}(\boldsymbol{v}) \setminus \{0\})$
8:     $\text{align}_{\boldsymbol{\beta}^2}(\boldsymbol{v}_0, \boldsymbol{v}) = (\text{response}(\boldsymbol{v}_0) \setminus \{0\}, 0)$
9: **end if**

---

Now, if $\mathsf{t} = +1$, then $(+1, 0)$ are aligned together (response of the same unknown vector) and $(\mathsf{s}, -1)$ are aligned together. Similarly, if $\mathsf{t} = -1$, then $(-1, 0)$ and $(+1, \mathsf{s})$ are aligned together respectively.

The alignment algorithm is presented in Algorithm 13. It makes $O(\log n)$ queries and succeeds with probability at least $1 - 1/n$.

---

**Algorithm 13** ALIGN QUERIES, CASE 2

**Require:** $\boldsymbol{v}_0, \boldsymbol{v} \in \{-1, 1\}^n$, $0 \in \text{response}(\boldsymbol{v})$, $\text{response}(\boldsymbol{v}_0) = \{\pm 1\}$.
1: Set batchsize $T = O(\log n)$.
2: Set $\mathsf{Inf}$ to be a large positive number.
3: Query$(\boldsymbol{v}_0 + \mathsf{Inf} \cdot \boldsymbol{v}, T)$.
4: **if** $\text{response}(\boldsymbol{v}_0 + \mathsf{Inf} \cdot \boldsymbol{v}) = \{\text{response}(\boldsymbol{v}) \setminus \{0\}, +1\}$ **then**
5:     $\text{align}_{\boldsymbol{\beta}^1}(\boldsymbol{v}_0, \boldsymbol{v}) = (+1, 0)$
6:     $\text{align}_{\boldsymbol{\beta}^2}(\boldsymbol{v}_0, \boldsymbol{v}) = (-1, \text{response}(\boldsymbol{v}) \setminus \{0\})$
7: **else**
8:     $\text{align}_{\boldsymbol{\beta}^1}(\boldsymbol{v}_0, \boldsymbol{v}) = (+1, \text{response}(\boldsymbol{v}) \setminus \{0\})$
9:     $\text{align}_{\boldsymbol{\beta}^2}(\boldsymbol{v}_0, \boldsymbol{v}) = (-1, 0)$
10: **end if**

$\square$

*Proof of Proposition 21.* The objective of Proposition 21 is to align the responses of queries $\boldsymbol{v}_0$ and $\boldsymbol{v}$ by identifying which among the following two hypotheses is true:

- $\mathbb{H}_1$ : The response of the unknown vectors with both the query vectors $\boldsymbol{v}_0$ and $\boldsymbol{v}$ is same. Since we fixed the $\text{sign}(\langle \boldsymbol{v}_0, \boldsymbol{\beta}^1 \rangle) = 1$, this corresponds to the case when $\text{align}_{\boldsymbol{\beta}^1}(\boldsymbol{v}_0, \boldsymbol{v}) = (+1, +1)$ and $\text{align}_{\boldsymbol{\beta}^1}(\boldsymbol{v}_0, \boldsymbol{v}) = (-1, -1)$.

  In this case, we observe that for any query of the form $\eta \boldsymbol{v}_0 + \zeta \boldsymbol{v}$ with $\eta, \zeta > 0$, the response set will remain $\{+1, -1\}$.

- $\mathbb{H}_2$ : The response of each unknown vector with both the query vectors $\boldsymbol{v}_0$ and $\boldsymbol{v}$ is different, i.e., $\text{align}_{\boldsymbol{\beta}^1}(\boldsymbol{v}_0, \boldsymbol{v}) = (+1, -1)$ and $\text{align}_{\boldsymbol{\beta}^1}(\boldsymbol{v}_0, \boldsymbol{v}) = (-1, +1)$.

  In this case, we note that the response for the queries of the form $\eta \boldsymbol{v}_0 + \zeta \boldsymbol{v}$ changes from $\{-1, 1\}$ to either $\{+1\}, \{-1\}$, or $\{0\}$ for an appropriate choice of $\eta, \zeta > 0$. In particular, the cardinality of the response set for queries of the form $\eta \boldsymbol{v}_0 + \zeta \boldsymbol{v}$ changes from 2 to 1 if $\frac{\eta}{\zeta} \in \left[ -\frac{\langle \boldsymbol{\beta}^1, \boldsymbol{v} \rangle}{\langle \boldsymbol{\beta}^1, \boldsymbol{v}_0 \rangle}, -\frac{\langle \boldsymbol{\beta}^2, \boldsymbol{v} \rangle}{\langle \boldsymbol{\beta}^2, \boldsymbol{v}_0 \rangle} \right] \cup \left[ -\frac{\langle \boldsymbol{\beta}^2, \boldsymbol{v} \rangle}{\langle \boldsymbol{\beta}^2, \boldsymbol{v}_0 \rangle}, -\frac{\langle \boldsymbol{\beta}^1, \boldsymbol{v} \rangle}{\langle \boldsymbol{\beta}^1, \boldsymbol{v}_0 \rangle} \right]$.

---

**Algorithm 14** ALIGN QUERIES, CASE 3

---

**Require:** $\boldsymbol{v}_0, \boldsymbol{v} \in \{0, -1, 1\}^n$, $\text{response}(\boldsymbol{v}) = \text{response}(\boldsymbol{v}_0) = \{\pm 1\}$.
1: Set batchsize $T = O(\log nk/\delta)$.
2: **for** $\eta \in \{\frac{c}{d} \mid c, d \in \mathbb{Z} \setminus \{0\}, |c|, |d| \le \frac{\sqrt{k}}{\delta}\}$ **do**
3:     Query$(\eta \boldsymbol{v}_0 + \boldsymbol{v}, T)$.
4:     **if** $|\text{response}(\eta \boldsymbol{v}_0 + \boldsymbol{v})| == 1$ **then**
5:        Return $\text{align}_{\boldsymbol{\beta}^1}(\boldsymbol{v}_0, \boldsymbol{v}) = (+1, -1)$, $\text{align}_{\boldsymbol{\beta}^2}(\boldsymbol{v}_0, \boldsymbol{v}) = (-1, +1)$
6:     **end if**
7: **end for**
8: Return $\text{align}_{\boldsymbol{\beta}^1}(\boldsymbol{v}_0, \boldsymbol{v}) = (+1, +1)$, $\text{align}_{\boldsymbol{\beta}^2}(\boldsymbol{v}_0, \boldsymbol{v}) = (-1, -1)$

---

In order to distinguish between these two hypotheses, Algorithm 14 makes sufficient queries of the form $\eta \boldsymbol{v}_0 + \zeta \boldsymbol{v}$ for varying values of $\eta, \zeta > 0$. If for some $\eta, \zeta$ the cardinality of the response set changes from 2 to 1, then we claim that $\mathbb{H}_2$ holds, otherwise $\mathbb{H}_1$ is true. Algorithm 14 then returns the appropriate alignment.

Note that for any query vector $\boldsymbol{v} \in \{-1, 1\}^n$, and any $k$-sparse unknown vector $\boldsymbol{\beta} \in \mathbb{S}^{n-1}$ the inner product $\langle \boldsymbol{\beta}, \boldsymbol{v} \rangle \in [-\sqrt{k}, \sqrt{k}]$. Moreover, if we assume that the unknown vectors have precision $\delta$, the ratio $\frac{\langle \boldsymbol{\beta}^2, \boldsymbol{v} \rangle}{\langle \boldsymbol{\beta}^2, \boldsymbol{v}_0 \rangle}$ can assume at most $4k/\delta^2$ distinct values. Algorithm 14 therefore iterates through all such possible values of $\eta/\zeta$ in order to decide which among the two hypothesis is true.

The total number of queries made by Algorithm 14 is therefore $4kT/\delta^2 = O(\frac{k}{\delta^2} \log(\frac{nk}{\delta}))$. From Lemma 5, all the responses are recovered correctly with probability $1 - O(1/n)$. $\square$

## E Experiments

Similar to the mixed regression model, the problem of learning mixed linear classifiers can be used to model heterogenous data with categorical labels. We provide some simulation results to show the efficacy of our proposed algorithms to reconstruct the component classifiers in the mixture.

Moreover, the algorithm suggested in this work can be used to learn the set of discriminative features of a group of people in a crowd sourcing model using simple queries with binary responses. Each person's preferences represents a sparse linear classifier, and the oracle queries here correspond to the crowdsourcing model. To exemplify this, we provide experimental results using the MovieLens [17] dataset to recover the movie genre preferences of two different users (that may use the same account, thus generating mixed responses) using small number of queries.

## E.1 Simulations

We perform simulations that recover the support of $\ell = 2$, $k$-sparse vectors in $\mathbb{R}^n$ using Algorithm 8. We use random sparse matrices with sufficient number of rows to construct an RUFF. Error is measured in terms of relative hamming distance between the actual and the reconstructed support vectors.

The simulations show an improvement in the accuracy with increasing number of rows allocated to construct the RUFF for different values of $n = 1000, 2000, 3000$ with fixed $k = 5$. This is evident since the increasing number of rows improve the probability of getting an RUFF.

Figure 2: Support Recovery for $\ell = 2, k = 5$ and $n = 1000, 2000, 3000$.

## E.2 Movie Lens

The MovieLens [17] database contains the user ratings for movies across various genres. Our goal in this set of experiments is to learn the movie genre preferences of two ($\ell = 2$) unknown users using a small set of commonly rated movies.

We first preprocess the set of all movies from the dataset to obtain a subset that have an average rating between 2.5 to 3.5. This is done to avoid biased data points that correspond to movies that are liked (or not liked at all) by almost everyone. For the rest of the experiment, we work with this pre-processed set of movies.

We consider $n = 20$ movie genres in some arbitrary, but predetermined order. The genre preference of each user $i$ is depicted as an (unknown) indicator vector $\boldsymbol{\beta}^i \in \{0, 1\}^n$, i.e., $\boldsymbol{\beta}_j^i = 1$ if and only if user $i$ likes the movies in genre $j$. We assume that a user *likes* a particular movie if they rate it 3 or above. Also, we assume that the user likes a genre if they like at least half the movies they rated in a particular genre.

We consider two users, say $U_1, U_2$ who have commonly rated at least 500 movies. The preference vectors for both the users is obtained using Algorithm 8. We query the *oracle* with a movie, and obtain its rating from one of the two users at random. For the algorithm, we consider each query to correspond to the indicator of genres that the queried movie belongs to. Using small number of such randomly chosen movie queries, we show that Algorithm 8 approximately recovers the movie genre preference of both the users.

First, we pick a random subset of $m$ movies that were rated by both the users, and partition them into two subsets of size $m_1$, and $m_2$ respectively. The first set of $m_1$ movies are used to partition the list of genres into three classes - genres liked by exactly one of the users, genres liked by both the users, and the genres liked by neither user. These set of $m_1$ randomly chosen movies essentially correspond to the rows of a RUFF used in Algorithm 8.

We then align the genres liked by exactly one of the users, we use the other set of $m_2$ randomly chosen movies and obtain two genre preference vectors $\boldsymbol{s_1}, \boldsymbol{s_2}$. Since we do not know whether $\boldsymbol{s_1}$ corresponds the preference vector of $U_1$ or $U_2$, we validate it against both, i.e., we validate $\boldsymbol{s_1}$ with $U_1$, $\boldsymbol{s_2}$ with $U_2$ and vice versa and select the permutation with higher average accuracy.

**Validation:** In order to validate our results, we use our recovered preference vectors to predict the movies that $U_1$ and $U_2$ will like. For each user $U_i$, we select the set of movies that were rated by $U_i$, but were not selected in the set of $m$ movies used to recover their preference vector. The accuracy

of our recovered preference vectors are measured by correctly predicting whether a user will like a particular movie from the test set.

**Results:** We obtain the accuracy, precision and recall for three random user pairs who have together rated at least 500 movies. The results show that our algorithm predicts the movie genre preferences of the user pair with high accuracy even with small $m$. Each of the quantities are obtained by averaging over 100 runs.

| id: $(U_1, U_2)$ | $m_1$ | $m_2$ | A($U_1$) | P($U_1$) | R($U_1$) | A($U_2$) | P($U_2$) | R($U_2$) |
|---|---|---|---|---|---|---|---|---|
| | 0 | 0 | 0.300 | 0.000 | 0.000 | 0.435 | 0.000 | 0.000 |
| (68, 448) | 10 | 20 | 0.670 | 0.704 | 0.916 | 0.528 | 0.550 | 0.706 |
| | 30 | 60 | 0.678 | 0.700 | 0.944 | 0.533 | 0.548 | 0.791 |
| | 0 | 0 | 0.269 | 0.000 | 0.000 | 0.107 | 0.000 | 0.000 |
| (274, 380) | 10 | 20 | 0.686 | 0.733 | 0.902 | 0.851 | 0.893 | 0.946 |
| | 30 | 60 | 0.729 | 0.737 | 0.982 | 0.872 | 0.891 | 0.976 |
| | 0 | 0 | 0.250 | 0.000 | 0.000 | 0.197 | 0.000 | 0.000 |
| (474, 606) | 10 | 20 | 0.665 | 0.752 | 0.827 | 0.762 | 0.804 | 0.930 |
| | 30 | 60 | 0.703 | 0.750 | 0.910 | 0.787 | 0.806 | 0.970 |