[Reviews · NeurIPS 2020]

Review 1

Summary and Contributions: This paper considers the problem of learning from a mixture of sparse classifiers. Specifically, the problem supposes that we have an ensemble of linear classifiers that each depend on a small subset of features. Each time we make an observation we provide a vector that is classified using one of these classifiers, but we do not know which one (it is randomly selected). The goal is to recover (the normal vector) for each of the classifiers from this mixture of observations. There has been some prior work on a related problem (in a regression context), but I think this represents the first work showing that recovering classifiers from this type of observation is even possible. [After reviewing the author's feedback I am increasing my rating slightly. While I am still hopeful that a stronger result may be possible (using less restrictive assumptions), I think the authors make a good case that this assumption is perhaps a bit less restrictive than I had initially thought. In any case, even with a strong assumption, I think this is an interesting and important first step in tackling this problem.]

Strengths: Overall I think this is an interesting problem, and the authors provide the first guarantees for this setting. While I beleive there may be room for improvement in terms of weakening the assumptions, this seems like a good first step. The theoretical approaches taken by the authors seem novel and interesting.

Weaknesses: The biggest limitations of this work are that: 1) I feel that their assumption 1 (essentially, that every classifier uses at leats one unique feature) seems overly restrictive. It is not obvious to me that this is strictly necessary. For example, consider 3 2-sparse classifiers that use features (1,2), (2,3), and (1,3). Is it really impossible to learn to distinguish these? 2) There are no simulations provided, so it is hard to say how this would work in practice.

Correctness: As far as I can tell.

Clarity: Overall the paper is written relatively clearly, but one feature which makes it hard to parse is that many key details were relegated to the supplemental material. In particular, key components of the algorithms (Algs 3 and 4) were not even discussed at all in the main body. If you only read the main body, you don't really have any intuition for how the main algorithm actually works aside from the fact that it splits the problem into two stages (support identification, and then estimation). I think it would be worth stating fewer theorem statements and formal definitions and providing a more thorough/intuitive discussion of how the algorithms actually work.

Relation to Prior Work: Yes.

Reproducibility: Yes

Additional Feedback:


Review 2

Summary and Contributions: This work initiates the study of the following generalization of 1-bit compressed sensing. There are some unknown k-sparse vectors w_1,...,w_{ell} in R^d, and one can query any vector v and get back sgn(<v,w_i>) for random index i. The goal is to recover the w_i's while minimizing the number of queries. This problem should not be confused with the problem of learning mixtures of halfspaces in the sense of distribution learning, as here the learner gets to pick the design vectors. A similar model in the context of regression has been studied before by Krishnamurthy et al. and Yin et al., as the authors acknowledge. For the case of general ell, they operate under the assumption that every w_i has some coordinate in its support which is not in the support of any other w_j. This assumption is directly inspired by the notion of "separability" in the nonnegative matrix factorization literature which has also manifested in the topic modeling literature under the name of "anchor word." Under this assumption, they give an algorithm with poly(ell,k,log n,1/eps) query complexity that can be made nonadaptive. For the special case of ell=2, they give an algorithm under milder assumptions. They also report some experimental results including both synthetic experiments and an experiment on the MovieLens dataset. In the latter, they used their algorithm to predict movie preferences for pairs of users, the premise being that a query vector corresponds to getting a rating for a specific movie from one of the two users chosen at random, and a user from the pair is chosen at random because they might be sharing accounts. The techniques draw upon the approach of Acharya-Bhattacharyya-Kamath for 1-bit compressed sensing. In that setting, the main challenge is to recover the support of the single underlying vector. A naive way to do this would be to query each of the standard basis vectors, but this is quite wasteful and the query complexity would end up depending on the dimension. Robust union-free families give an elegant way of doing this more efficiently, and the work of Acharya-Bhattacharyya-Kamath leverage these set systems to get better query complexity for 1-bit compressed sensing. The present work uses the same insight to identify for every coordinate i the number of vectors in the mixture with support containing i. A generalization of RUFFs can also be leveraged to estimate for pairs of coordinates i,j the number of vectors with support containing both i and j. Altogether, this gives one access to the entries of XX^T, where X is the Boolean matrix whose columns are indicators for the supports of the mixture components, and then using the separability assumption, they can uniquely recover X by a constrained matrix factorization (a la Arora-Ge-Kannan-Moitra). ---------------- Update: Thanks for the clarifications! I'm happy to walk back my claim about the large polynomial dependence on k,l, and I'm upgrading my score to a 6. Regarding the other results proven in the supplementary material that the rebuttal mentioned, the l=2 result does seem exciting in that, in the absence of the strong separability assumption, one needs to handle the case where the supports of the two vectors are identical, so eps-recovery requires new ideas (e.g. Proposition 21 seems to be the most delicate case to handle in the alignment procedure). The resulting algorithm is interesting but admittedly quite specific to l=2.

Strengths: The primary strength is to introduce this particular problem, which seems like a fairly natural and well-motivated question and more challenging than its regression analogue.

Weaknesses: Given that this is primarily a theoretical work, I'm not sure if the results are sufficiently technically interesting to merit acceptance. One issue is that the assumption of separability seems rather artificial. Even in the setting of the MovieLens experiment, if a group of ell users were queried, it would be quite strange if every user liked some genre that no other user in the group liked. Another issue is that the query complexity is a rather large polynomial in k,ell, and the paper doesn't explore what the right dependence on k, ell should be in terms of lower bounds. Admittedly, the use of RUFFs is nice and it takes some extra work to make this approach work beyond the traditional 1-bit compressed sensing setup, but overall the results in the present work feel a bit on the preliminary side.

Correctness: I have verified that the proofs are correct.

Clarity: The writing is fine, though one minor complaint is that Sections 2 and 3 could be better integrated into Section 1 to avoid stating the results/model multiple times.

Relation to Prior Work: The relation to prior work is clearly discussed.

Reproducibility: Yes

Additional Feedback: Is there any hope of handling the case where the mixing weights are non-uniform and unknown with these techniques? Minor typos: - There are a few places where you say "compress sensing"


Review 3

Summary and Contributions: The paper considers a problem of recovering multiple sparse vectors via oracle access. Suppose there are \ell k-sparse vectors v_1,...,v_\ell in R^n which are unknown to the algorithm. But the algorithm has access to an oracle, which, when queried on an arbitrary vector x in R^n, chooses i in {1,...,\ell} uniformly at random and returns sgn(<x,v_i>). The goal of the algorithm is to recover v_1,...,v_\ell (up to a common permutation of the coordinates) with as few oracle queries as possible. When \ell = 1, this is the 1-bit compressive sensing, which is now well-understood. When \ell = 2, assuming the sparse vectors have disjoint supports, existing results show O(k^3 log^2 n) queries suffices. This paper shows that for a general \ell, under the assumption that the support of any sparse vector is not contained in the union of the supports of other sparse vectors, O(\ell^6 k^3 log^2 n) queries suffices. The key step is to recover the support of those sparse vectors. By repeating the queries on the same x, we can recover the number of sparse vectors such that <x,v_i> is nonzero. The problem then carries a strong flavour of combinatorial design/group testing. The recovery algorithm is then based on some previous combinatorial designs called union free families and cover free families. After finding the supports, one can use an additional O(\ell^3 k polylog(nk)) Gaussian queries to recover the sparse vectors. Here, one first determines the sign of the coordinate of each sparse vector that is not contained in the support of other sparse vectors and then use a clever design to find the sgn(<g,v_i>) for each i for a Gaussian vector g (I really like this part), which reduces to the 1-bit compressive sensing for v_i.

Strengths: This work is a nontrivial extension of a well-studied problem to the case of multiple sparse vectors. It is a solid contribution and the techniques are potentially useful to other problems such as in group testing. It will be of interest to a good fraction of the NeurIPS community, including those who work on compressive sensing and group testing.

Weaknesses: [I read the rebuttal, in which the authors had adequately addressed my concerns.]

Correctness: I did not verify all the details but the algorithms are initutively correct.

Clarity: The paper is very well-written. However, it would be good to have a technique overview (even in the supplementary). Line 116: 1-bit compress sensing -> 1-bit compressed sensing

Relation to Prior Work: Comparison with previous works in techniques looks inadequate to me. For instance, Algorithm 3 is similar in spirit to that in [1] (for one-bit compressive sensing). Some high-level description of the extension should be included.

Reproducibility: Yes

Additional Feedback:


Review 4

Summary and Contributions: The authors study the problem of learning sparse linear classifiers from a mixture of responses. The authors provide rigorous theoretical arguements for the same problem. The authors provide analyses for two approaches: (a) a two-stage approach that first recovers the supports of the $l$ unknown sparse classifiers, followed by estimating the magnitudes of the non-zero elements; and (b) a non-adaptive approach wherein the classifiers are learnt using a one-shot approach but that comes at an increased query complexity, and a higher probability of failure.

Strengths: The theoretical contribution of this paper is quite solid, and the theory matches known results for the $l=1$ case. The analysis is fairly non-trivial and I believe this paper is a good addition to the ML community. To my knowledge, there is no existing work that provides theoretical guarantees for this problem, and thus it is an excellent first step towards understanding this problem.

Weaknesses: The upper bound on the query complexity is a bit hard to parse. I understand that this problem is inherently harder than the classical 1-bit compressive sensing problem, but the 6-th power dependence on l, and the cubic dependence on $k$ seems to be an artifact of the current analysis. A brief note explaining why this arises and/or any possible tricks to reduce this might make the paper better to read. In the same vein, the "exponential" dependence of the one-shot approach seems weak to me. Assumption (2) seems a bit too restrictive to me. And in fact, this seems to contradict the statement the authors make in the two-stage algorithm where they claim that the the magnitude of the non-zeros needs to be LOWER bounded by 1/poly(n). This requires some clarification.

Correctness: The arguments appear to be correct although I did not expressly check all the proofs.

Clarity: Overall the paper is well written, and quite easy to follow. There are some notational issues: (a)what is \delta \Z^n? (b) what is the batchsize in Algorithm 1? Is there a regime in which the Chernoff bounds make more sense to apply? (c) I find it a bit strange that the experimental section is completely relegated to the appendix.

Relation to Prior Work: The work is placed well within the context of prior work and to the best of my knowledge, there are no obvious references that are missing.

Reproducibility: Yes

Additional Feedback: I thank the authors for their interesting response.

[Author Response · NeurIPS 2020]

We thank the reviewers for their comments on the manuscript. We address a point about the separability assumption
and the polynomial dependency below before giving individual responses subsequently.

**Separability assumption**: Note that the separability assumption is used only for the theoretical analysis; the algorithms
will run and produce results even without it. To make first theoretical forays in similar setting, it is common to use
idealized assumptions that give further insights. Note that, the separability assumption of the supports is not very
unnatural and similar assumptions have been made for non-negative integer matrix factorization [3,12,21], clustering
(Huleihel et. al. 2019) etc. Indeed, when the number of features is large and fine-grained, and $\ell$ is small, this condition
is often satisfied. Intuitively, if the non-zero indices of the unknown sparse vectors are chosen uniformly at random
for each unknown vector, then with high probability the separability assumption is true. As mentioned in L123-L141,
the separability assumption is used for both the support recovery as well as the $\epsilon$-recovery objective. Although the
separability assumption can perhaps be relaxed for the support recovery, it seems unlikely that it can be relaxed for the
$\epsilon$-recovery objective.
**Polynomial dependence on $k$ and $\ell$**: Even to recover the support of a single $k$-sparse vector in the 1-bit CS setting,
$\tilde{\Omega}(k^2)$ queries are required [1]. In comparison we use $\tilde{O}(k^3)$ queries for the support recovery of multiple vectors. Here
a cubic dependence on $k$ is not very unreasonable. Note that, the cubic dependence stems from the usage of the pairwise
union free families that have about $k^3$ rows and are optimal. The dependence on $\ell$ is large, while the lower bound must
grow at least linearly with $\ell$. We agree that finding the correct lower bounds for general $\ell$ is an interesting open problem.
For now there exists this rather large gap between the upper and lower bounds. However, $\ell$ is generally considered to
be a constant in similar mixture models (see [23],[34]). Also, note that an exponential dependence on $\ell$ is sometimes
unavoidable for results on sample complexities of Mixtures (see Moitra and Valiant, 2010). In this work, this large
dependence on $\ell$ stems from the usage of generalized union free families.

*Response to Reviewer 1*: **"Necessity of separability"** Please see the discussion above regarding the assumptions. Note
that, the assumptions are sufficient for the recovery of the vectors and are not necessary. In the example provided it
is possible to recover the support of the unknown vectors since the corresponding gram matrix $XX^T$ has this unique
square-root up to a permutation. This might not always be the case. Also note that, for your example the 'complement'
(0s and 1s swapped) matrix is separable, which makes the gram matrix to factorize uniquely. All this is just for support
recovery, for $\epsilon$-recovery our analysis also uses separability assumption crucially (cf. Algorithms 5,7 in supplementary).
**"Simulations"** Section E of the supplementary material contains some experiments on both synthetic and real world
data. Since this work is mostly theoretical, we have relegated the experimental sections to the appendix.

*Response to Reviewer 2*: Apart from the summary the reviewer provided, we would like to point out that there are
rather intricate results, such as, $\epsilon$ recovery of distinct classifiers for general $\ell$, a completely non-adaptive algorithm for $\epsilon$
recovery, and also the recovery of 2 classifiers without the separability assumptions, that are not stated in the summary.
While much of those proofs are in the supplementary, we urge the reviewer to take those into consideration as well. The
major challenges in this work come from the de-mixing and alignment tasks, and the techniques used in this work go
beyond just the usage of 1-bit compressed sensing methods, so we do not think of this as an extension of 1-bit CS.
**"Separability"** Please refer to the discussion on assumptions above. In the MovieLens example, genres are too crude
features. In most cases more fine-grained features are used, and $\ell$ is small, which make the assumption more plausible.
For support recovery algorithm (Algorithm 2), the separability assumption can be relaxed slightly. Moreover, for $\ell = 2$,
such an assumption is not required even for $\epsilon$ recovery.
**"Complexities are large polynomials"** Please see the polynomial dependence point above.
**"Nonuniform mixtures"** The case of non-uniform mixing weights is an interesting question. For the case of $\ell = 2$
(two unknown vectors), the exact mixing weights need not be known and we just need to know a lower bound on
the smallest weight. For larger $\ell$ the same technique does not extend directly and will require non-trivial methods to
estimate the *counts*.

*Response to Reviewer 3:* Thanks for appreciating the work and the constructive feedback. **"Experiments with $\ell = 2$"**
Even for $\ell = 2$ case, our results are the first ones (Thm. 4, cor. 1, and the $\ell = 2$ cases of other Theorems), and therefore,
no comparative results exists. Note that, [31] does not provide results of this flavor so not directly comparable.

*Response to Reviewer 4:* **"Dependence on $k$ and $\ell$"** Please see the polynomial dependence point above. We will
elaborate this in the introduction as suggested.
**"Assumption 2"** This assumption is required for the correctness of Thm 4; whereas the lower bound on the magnitudes
of the non-zero entries is required to determine an upper bound on value for Inf with a polynomial bit-complexity and
not at all required for the correctness. That being said, both the conditions can be simultaneously satisfied as well (an
element can be $o(1)$ but greater than $1/poly(n)$). We will add a discussion to clarify this in the final version.
**"Clarity"** (a) $\delta\mathbb{Z}^n$ refers to the set of $n$-dimensional vectors with entries being integer multiple of $\delta$.
(b) The batchsize is defined in line 259 of the manuscript. We can apply Chernoff-type bounds here since the oracle
answers all queries independently.

[Meta-Review · NeurIPS 2020]

The reviewers uniformly felt that this is an interesting paper, and a good contribution to the community.